# Refinement Methods for Distributed Distribution Estimation under $\ell^p$-Losses

**Deheng Yuan**[1]**, Tao Guo**[2,3]**, Zhongyi Huang**[1]
[1]Department of Mathematical Sciences, Tsinghua University
[2]School of Cyber Science and Engineering, Southeast University
[3]State Key Laboratory of Integrated Services Networks, Xidian University
`ydh22@mails.tsinghua.edu.cn, taoguo@seu.edu.cn,`
`zhongyih@tsinghua.edu.cn`

## Abstract

Consider the communication-constrained estimation of discrete distributions under $\ell^p$ losses, where each distributed terminal holds multiple independent samples and uses limited number of bits to describe the samples. We obtain the minimax optimal rates of the problem for most parameter regimes. As a result, an elbow effect of the optimal rates at $p = 2$ is clearly identified. In order to achieve the optimal rates for different parameter regimes, we introduce refinement methods and develop additional customized techniques in the estimation protocols. The general idea of the refinement methods is to first generate rough estimate by partial information and then establish refined estimate in subsequent steps guided by the rough estimate. Then customized techniques such as successive refinement, sample compression, thresholding and random hashing are leveraged to achieve the optimal rates in different parameter regimes. The optimality of the estimation protocols is shown by deriving compatible minimax lower bounds.

## 1 Introduction

Motivated by applications in areas such as federated learning [1–3], distributed statistical estimation problems have recently received wide attention. In this setting, multiple distributed agents cooperate to train a model, while each of them can only access to a subset of training data. These agents can exchange messages but their communication budgets are constrained. The performance of the system is often limited by the communication constraints.

One fundamental learning task is to estimate the underlying discrete distribution of the data. Under communication constraints, the minimax optimal rates for the estimation error were studied in [4–11]. Another important constraint is the differential privacy, and the corresponding problem was similarly considered in [5,6,12,13]. In these works, only one sample was accessed by each distributed terminal and the most common $\ell^1$ and $\ell^2$ losses were used to measure the estimation error. However, this is an oversimplification of the practical case, where general $\ell^p$ losses may be necessary and each terminal can access to $n > 1$ samples.

On the one hand, some works [14,15] further explored the distribution estimation problem with $n > 1$ samples at each terminal, under the $\ell^1$ loss. On the other hand, later works [16,17] considered the problem under general $\ell^p$ losses, with a limited scope to $n = 1$. Even in this limited case, for the regime $p > 2$ only suboptimal lower bounds were derived. In the more practical case where each terminal can obtain $n > 1$ samples, the optimal rates under $\ell^p$ losses are also unclear. The problem with $n > 1$ samples is much more difficult than that for $n = 1$, since its inherent structure is not

---

The first two authors contributed equally to this work. Corresponding authors: Tao Guo, Zhongyi Huang.

39th Conference on Neural Information Processing Systems (NeurIPS 2025).

revealed in the $n = 1$ case. Even though [14] presented an optimal protocol for $n > 1$ and the $\ell^1$ loss, it still does not apply to $\ell^p$ losses since its optimality depends heavily on several special properties of the $\ell^1$ loss.

In this work, we consider the distributed estimation of discrete distributions under communication constraints. The range of the problem is expanded in two directions, letting each terminal hold $n > 1$ samples and imposing general $\ell^p$ losses simultaneously. We design interactive protocols to achieve optimal rates in this technically more challenging setting. The difficulty lies in the communication budget allocation strategy, namely how to assign multiple terminals and their communication budgets to the tasks of estimating different distribution entries. The naive uniform allocation strategy that treats all the entries equally fails to achieve the optimal convergence rate under the general $\ell^p$ loss for $p > 1$. To achieve the optimal rate, communication budgets should be invested based on the distribution to be estimated. As a result, existing protocols cannot handle the general problem under the $\ell^p$ loss. Instead, we develop refinement methods in the estimation protocol, which first establishes rough estimate based on partial information obtained by a portion of budgets, and then uses it to allocate the remaining budgets for refining the estimate. The refined estimate can achieve the optimal error rate, since the remaining budgets are allocated most effectively.

We introduce additional auxiliary estimation techniques to customize the refinement methods for different parameter regimes. The induced estimation protocols shows upper bounds for the optimal rates. We also derive compatible lower bounds for most parameter regimes. Hence the optimality of the protocols is shown and the optimal rates are obtained in these regimes.

1. We exploit the classic divide-and-conquer technique and design a successive refinement estimation protocol equipped with an adaptive budget allocation strategy. The distribution is divided into blocks. The estimation task is achieved by first estimating the block distribution and then conditional distribution over each block. The block distribution has a lower dimension, and the divide-and-conquer procedure is not stopped until it is more efficient to estimate each entry directly. This induces a successive refinement protocol where the rough estimate for the block distribution is refined by further estimating the conditional distributions over blocks. More importantly, in the refinement step we introduce an adaptive budget allocation strategy. Specifically, terminals are assigned to estimating different conditional distributions based on the block distribution estimated by the former phase, which achieves faster convergence rate for $p > 1$ than the uniform allocation strategy by previous works [14]. Hence the successive refinement protocol achieves the optimal rates up to logarithmic factors for most parameter regimes with $1 \leq p \leq 2$. Moreover, by using multiple successive refinement steps rather than only one step, our protocol for $p = 1$ achieves the optimal rates for a larger range of regimes than that in [14].

2. For $p > 2$, we develop auxiliary sample compression techniques, so that refinement methods can be adopted in the estimation protocol. Different from $1 \leq p \leq 2$, the protocol in this regime obtains a rough estimate of the distribution itself (rather than an estimate of the block distribution) first by uniform allocation of budgets. It then refines the estimate by allocating the remaining budgets according to the rough estimate. In the refinement stage of the protocol, we further develop sample compression techniques, which compress the description for samples and reduce the communication budget, allowing more samples to be transmitted. The resulting protocols can achieve the optimal rates for relatively large $n$.

3. In the very special regime where the total communication budget is extremely tight, we incorporate a thresholding technique into the estimation protocol to achieve the optimal rate. The key observation is that under the extremely tight communication budget, if an entry of the distribution is too small then approximating it simply by 0 induces a lower variance than trying to estimate them. For $p > 2$, the thresholding technique are combined with the sample compression to yield the optimality protocol. To the best of our knowledge, the regime has not been discussed in any previous work.

4. For the special case $n = 1$, we design an optimal non-interactive protocol by exploiting random hash functions, rather than the sample splitting trick or the simulate and infer protocol used in previous works [7, 8, 18]. To show the optimality, we further establish a compatible lower bound that is strictly better than that in [16, 17] for $p > 2$. This proves the optimal rates under general $\ell^p$ losses, especially that for $p > 2$ left open by previous works [16, 17].

The expression of the optimal rates under $\ell^p$ losses reveals an elbow effect at $p = 2$, providing more insights into the distributed estimation problem. It is interesting to compare our results with the elbow effect discovered in the nonparamentric density estimation problem [19, 20]. It is not a coincidence since in both problems there are constraints for the estimated object (namely the normalization constraint for the distribution estimation problem and the Sobolev regularity constraint for the nonparametric density estimation problem), and the loss functions can vary with a parameter. The similarity sheds light on how the optimal rates are affected by the relation between the imposed loss function and the constraints on the estimated object.

The remaining part of this work is organized as follows. First, the problem is formulated in Section 2. Then we present our main results for $1 \leq p \leq 2$ and $p > 2$ in Sections 3 and 4 respectively. In Section 5, the special case with $n = 1$ is discussed and the non-interactive protocol is presented. Finally, the optimal rates are summarized in Section 6 and a few further remarks are given in Section 7. Detailed estimation protocols and complete proofs of both upper and lower bounds can be found in the technical appendix.

## 2 Problem Formulation

Denote a discrete random variable by a capital letter and its finite alphabet by the corresponding calligraphic letter, e.g., $W \in \mathcal{W}$. We use the superscript $n$ to denote an $n$-sequence, e.g., $W^n = (W_i)_{i=1}^n$. For a finite set $\mathcal{W}$ of size $k = |\mathcal{W}|$, let $\Delta_{\mathcal{W}}$ be the set of all the probability measures over $\mathcal{W}$, i.e. $\Delta_{\mathcal{W}} \triangleq \{p(\cdot) : p(w) \in [0,1], \forall w \in \mathcal{W}, \sum_w p(w) = 1\}$. Let $\Delta'_{\mathcal{W}}$ be the set of subprobability measures, i.e. $\Delta'_{\mathcal{W}} \triangleq \{p(\cdot) : p(w) \in [0,1], \forall w \in \mathcal{W}, \sum_w p(w) \leq 1\}$.

Suppose that we want to estimate the finite-dimensional distribution $p_W \in \Delta_{\mathcal{W}}$ with dimension $k$, and the samples are generated at random. To be precise, let $W_{ij} \sim p_W(w), i = 1, 2, \cdots, m$, $j = 1, 2, \cdots, n$ be i.i.d. random variables distributed over $\mathcal{W}$. The total sample size is $mn$.

Consider the distributed minimax parametric distribution estimation problem with communication constraints depicted in Fig. 1, which is a theoretical model of federated learning systems. There are $m$ encoders and one decoder, and common randomness is shared among them. The $i$-th encoder observes the samples $W_i^n = (W_{ij})_{j=1}^n$ and transmits an encoded message $B_i$ of length $l$ to the decoder, $i = 1, ..., m$. Upon receiving messages $B^m = (B_i)_{i=1}^m$, the decoder needs to establish a reconstruction $\hat{p}_W \in \Delta'_{\mathcal{W}}$ of $p_W$.

An $(m, n, k, l)$-protocol $\mathcal{P}$ is defined by a series of random encoding functions

$$\text{Enc}_i : \mathcal{W}^n \times \{0,1\}^{(i-1)l} \to \{0,1\}^l, \forall i = 1, ..., m,$$

and a random decoding function

$$\text{Dec} : \{0,1\}^{ml} \to \Delta'_{\mathcal{W}}.$$

The $i$-th encoder is aware of the messages sent by the previous $i-1$ encoders (which can be achieved by interacting with other encoders and/or the decoder), and it generates a binary sequence $B_i = \text{Enc}_i(W^n, B_{1:i-1})$ of length $l$. The reconstruction of the distribution is $\hat{\boldsymbol{p}}_W^{\mathcal{P}} = \text{Dec}(B_1, B_2, ..., B_m)$.

For $p \geq 1$, we use the $\ell^p$ loss to measure the estimation error. We are interested in the minimal error of all the estimation protocols in the worst case, as the true distribution $p_W$ varies in the probability simplex $\Delta_{\mathcal{W}}$. To be specific, our goal is to characterize the order of the the following minimax convergence rate

$$R(m, n, k, l, p) = \inf_{(m,\,n,\,k,\,l)\text{-protocol } \mathcal{P}} \sup_{\boldsymbol{p}_W \in \Delta_{\mathcal{W}}} \mathbb{E}[\|\hat{\boldsymbol{p}}_W^{\mathcal{P}} - \boldsymbol{p}_W\|_p^p].$$

*Remark* 1. The $(m, n, k, l)$-protocol $\mathcal{P}$ defined in this work is usually called the (sequentially) interactive protocol in the literature. The protocol is called non-interactive, if for each $i = 1, ..., m$, the $i$-th encoder is ignorant of all the messages $B_{1:i-1}$ sent by previous encoders and the encoding function $\text{Enc}_i(W^n)$ is a function of the samples only. In most cases we design interactive protocols since it is too hard to construct a non-interactive protocol. For some simple special cases, non-interactive protocol achieving the optimal rates can be constructed, which will be indicated.

We further define some necessary notations. For any positive functions $a(m, n, k, l, p)$ and $b(m, n, k, l, p)$, we say $a \preceq b$ if $a \leq c \cdot b$ for some positive constant $c > 0$ independent of parameters $(m, n, k, l)$. The notation $\succeq$ is defined similarly. Then we denote by $a \asymp b$ if both $a \preceq b$ and $a \succeq b$ hold. Denote by $a \wedge b$ the minimum of two real numbers $a$ and $b$, and $a \vee b$ the maximum.

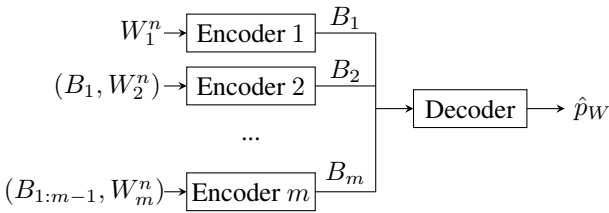

Figure 1: Distributed (sequentially) interactive distribution estimation

## 3    Optimal Rates for $1 \leq p \leq 2$

First assume that $1 \leq p \leq 2$. We present the upper bound in the following theorem.

**Theorem 1.** *Let $1 \leq p \leq 2$, then we have  $R(m,n,k,l,p) \preceq$*

$$
\begin{cases}
\dfrac{k}{(mnl)^{\frac{p}{2}}} \vee \dfrac{k^{1-\frac{p}{2}}}{(mn)^{\frac{p}{2}}}, & n \geq k,\, m(l \wedge k) > 1000k \log(mn) \log n, & \text{(1a)} \\[3ex]
\dfrac{k^{1-\frac{p}{2}} \log^{\frac{p}{2}}(\frac{k}{n}+1)}{(ml)^{\frac{p}{2}}} \vee \dfrac{k^{1-\frac{p}{2}}}{(mn)^{\frac{p}{2}}}, & \dfrac{k}{2^l} \leq n < k,\, m(l \wedge n) > 2000n \log(mn) \log n, & \text{(1b)} \\[3ex]
\dfrac{k}{(mn2^l)^{\frac{p}{2}}}, & n < \dfrac{k}{2^l},\, m(l \wedge n) > 4000n \log(mn) \log n, & \text{(1c)} \\[3ex]
\dfrac{\log^{\frac{p}{2}} k}{(ml)^{p-1}}, & \log k < l < n,\, ml < k. & \text{(1d)}
\end{cases}
$$

*Proof.* The case (1a) is by Proposition 1 in Appendix A, cases (1b) and (1c) are by Proposition 2 in Appendix B, and the case (1d) is by Proposition 5 in Appendix E. We sketch the proof here and details can be found in the appendix.

**Successive refinement protocol with adaptive budget allocation**    For the first three cases (1a), (1b) and (1c), the estimation protocol can be sketched as the following inductive procedure.

At each step, choose some $l_0$ and construct a division $\mathcal{W} = \cup_{s=1}^{t} \mathcal{W}_s$ with $|\mathcal{W}_s| \leq 2^{l_0} - 1$, $l_0 \leq l$ and $t = \lceil \frac{k}{2^{l_0}-1} \rceil$. First suppose that the distribution $\boldsymbol{p}_B$ of blocks has been estimated to some accuracy by some $\hat{\boldsymbol{p}}_B$. Then each encoder can use its $l_0$-bit message to only describe its samples on a predetermined block $\mathcal{W}_s$. Based on these messages, the decoder then estimates the conditional distribution $\boldsymbol{p}_s \in \Delta_{\mathcal{W}_s}$ on the $s$-th block, where $p_s(w) \triangleq p(w|\mathcal{W}_s)$. Based on the message, the decoder constructs $\hat{\boldsymbol{p}}_s$ as an estimate of $\boldsymbol{p}_s$. Combining $\hat{\boldsymbol{p}}_B$ and $\hat{\boldsymbol{p}}_s$ for each block $s$, an estimate of $p_W$ can be immediately obtained by letting $p_W(w) = \hat{p}_B(s)\hat{p}_s(w)$ for $w \in \mathcal{W}_s$. Note that $\boldsymbol{p}_B \in \Delta_{[1:t]}$ always has a lower dimension $t$ than the dimension $k$ for $p_W$. Fewer encoders are needed for the smaller problem. Hence $\hat{\boldsymbol{p}}_W$ can be refined from these layered block distributions successively.

For the base case $k < n$ where the length $k$ of the distribution $\boldsymbol{p}_W$ is sufficiently small, it is optimal to estimate $p_W(w)$ directly for each $w \in \mathcal{W}$ using the one-bit protocol in [14]. Although the error analysis is only shown for the $\ell^1$ loss in [14], it can be adapted to prove the optimality of the above procedure for $\ell^p$ losses with $1 \leq p \leq 2$. See Appendix A for details.

Then consider the successive refinement subroutine for estimating all the $\boldsymbol{p}_s$, $s = 1, ..., t$ given an estimate $\hat{\boldsymbol{p}}_B$ for $\boldsymbol{p}_B$. Through detailed error analysis (see Lemma 5 and its discussions), to achieve the optimality, the budget for estimating each $\boldsymbol{p}_s$ should be proportional to $\hat{p}_B(s)$. Since the decoder have obtained the rough estimate $\hat{p}_B(s)$, it can allocate remaining budget by interactions with encoders. Such an allocation plan is in contrast to the estimation problem under the TV loss discussed in Section 3.1 and [14], where a uniform budget allocation among all the $\boldsymbol{p}_s$, $s = 1, ..., t$ is optimal.

Given the successive refinement subroutine and the estimation protocol for the base case, it remains to consider the choice of $l_0$ as well as the budget allocation between these successive steps. They depend on the parameter regime. For the case (1b) where the length $l$ is relatively large, each message

can be divided to describe multiple samples. In order to directly exploit the protocol for the base case in estimating the block distribution $\boldsymbol{p}_B$, we choose $t \sim n$ and $l_0 \sim \log \frac{k}{n}$. In contrast, for the case (1c) where $l$ is relatively small, we let $l_0 = l$ and then use multiple successive steps until the dimension of the distribution is reduced to $n \cdot 2^l$. The complete estimation protocol is constructed in Appendices B.2.2 and B.2.3 respectively.

**Refinement protocol with thresholding techniques** For the final case (1d), the refinement protocol is designed with the help of thresholding techniques. The idea is that under the extremely tight communication budget, roughly $\sim ml$ samples can be transmitted by the protocol. Then approximating those $p_W(w) \preceq \frac{1}{ml}$ simply by 0 is better than estimating them. In the refinement step, the remaining budget can be used for generating another independent estimate for those $p_W(w) \succeq \frac{1}{ml}$, whose number $\sim ml$ is limited. Detailed protocol can be found in Appendix E.1. □

Lower bounds under the $\ell^p$ loss can be derived in the following lemma, which provides a baseline.

**Lemma 1.** *For $1 \leq p \leq 2$, we have*

$$
R(m,n,k,l,p) \succeq
\begin{cases}
\dfrac{k}{(mnl)^{\frac{p}{2}}} \vee \dfrac{k^{1-\frac{p}{2}}}{(mn)^{\frac{p}{2}}}, & n \geq k \log k, \, m > \left(\dfrac{k}{l}\right)^2 \\[2ex]
\dfrac{k^{1-\frac{p}{2}}}{(ml \log k)^{\frac{p}{2}}} \vee \dfrac{k^{1-\frac{p}{2}}}{(mn)^{\frac{p}{2}}}, & \dfrac{k}{2^l} \leq n < k \log k, \, m > \left(\dfrac{k}{l}\right)^2, \\[2ex]
\dfrac{k}{(mn2^l)^{\frac{p}{2}}}, & n < \dfrac{k}{2^l}, \, mn2^l > k^2, \\[2ex]
\dfrac{1}{(ml)^{p-1}} \vee \dfrac{k^{1-\frac{p}{2}}}{(mn)^{\frac{p}{2}}}, & ml < \dfrac{k}{2}.
\end{cases}
$$

*Proof.* Lower bounds for the first three cases under the $\ell^p$ loss can be derived from existing results in [14] under the $\ell^1$ loss. For the last case, we use an algebraic technique to first note $R(m,n,k,l,p) \geq R(m,n,2ml,l,p)$ and then bound the latter, which slightly strengthens the usual bound obtained by the data processing inequality. This induces compatible lower bound with the upper bound in (1d). The detailed proof can be found in Appendix G. □

Combining Theorem 1 and lemma 1, the optimal rates for the following cases can be roughly characterized by

$$
R(m,n,k,l,p) \asymp
\begin{cases}
\dfrac{k}{(mnl)^{\frac{p}{2}}} \vee \dfrac{k^{1-\frac{p}{2}}}{(mn)^{\frac{p}{2}}}, & n \geq k, \, ml \succeq k, \\[2ex]
\dfrac{k^{1-\frac{p}{2}}}{(ml)^{\frac{p}{2}}} \vee \dfrac{k^{1-\frac{p}{2}}}{(mn)^{\frac{p}{2}}}, & \dfrac{k}{2^l} \leq n < k, \, ml \succeq k, \\[2ex]
\dfrac{k}{(mn2^l)^{\frac{p}{2}}}, & n < \dfrac{k}{2^l}, \, mn2^l \succeq k^2, \\[2ex]
\dfrac{1}{(ml)^{p-1}} \vee \dfrac{k^{1-\frac{p}{2}}}{(mn)^{\frac{p}{2}}}, & ml \preceq k.
\end{cases}
\tag{2}
$$

*Remark* 2. We add a few explanations concerning the boundaries in (2). The regularity condition $m > (\frac{k}{l})^2$ in the lower bound is induced mainly by technical reasons and the boundary $ml > k$ is more essential. Similarly, the conditions $m(l \wedge k) > 1000k \log(mn) \log n$ and $m(l \wedge n) > 2000n \log(mn) \log n$ in the upper bound can be relaxed by finer analysis and the true boundaries seem to be around $ml > k$ and $ml > n$, respectively. Under these observations, in the third case the conditions $mn2^l \geq k^2$ and $n < \frac{k}{2^l}$ imply that $m > k$ and hence $ml > k > n$ is fullfilled.

### 3.1 Special Cases: Optimal Rates for $p = 1$ and $p = 2$

In this subsection, we specialize our results and characterize the optimal rates under the most commonly used total variation (TV) and squared losses, i.e. $\ell^1$ and $\ell^2$ losses. For the TV loss, the successive refinement protocol can be made non-interactive. See Appendix C for details.

**Theorem 2.** *The following upper bound can be achieved by a non-interactive protocol.*

$$R(m,n,k,l,1) \preceq \begin{cases} \sqrt{\dfrac{k^2}{mnl}} \vee \sqrt{\dfrac{k}{mn}}, & n \geq k, m(l \wedge k) > 1000k \log m \log n, \\[3mm] \sqrt{\dfrac{k\log(\frac{k}{n}+1)}{ml}} \vee \sqrt{\dfrac{k}{mn}}, & \dfrac{k}{2^l} \leq n < k, m(l \wedge n) > 2000n \log m \log n, \\[3mm] \sqrt{\dfrac{k^2}{mn2^l}}, & n < \dfrac{k}{2^l}, m(l \wedge n) > 4000n \log m \log n. \end{cases}$$

For the TV loss, we have the following characterization of the optimal rates.

$$R(m,n,k,l,p=1) \asymp \begin{cases} \sqrt{\dfrac{k^2}{mnl}} \vee \sqrt{\dfrac{k}{mn}}, & n \geq k, ml \succeq k, \\[3mm] \sqrt{\dfrac{k}{ml}} \vee \sqrt{\dfrac{k}{mn}}, & \dfrac{k}{2^l} \leq n < k, \ ml \succeq k, \\[3mm] \sqrt{\dfrac{k^2}{mn2^l}} \wedge 1, & n < \dfrac{k}{2^l}, \\[3mm] 1, & ml \preceq k. \end{cases} \tag{3}$$

*Remark* 3. In [14], a non-iterative protocol for the same problem in Section 2 under the TV loss is also constructed. However, corresponding to the third case in Theorem 2, in [14] a stronger restriction $m > 100\frac{k}{2^l}\log m \log n$ is imposed (cf. Theorem 1.1 in [14] and note that the notations $m$ and $n$ are interchanged therein). The restriction is induced by using the first bit of each encoder to estimate the block probability $\boldsymbol{p}_B$ with the protocol for the first case. The conditional probability in each block $B$ is then estimated. Combining it with the estimate for $\boldsymbol{p}_B$, an estimate for $\boldsymbol{p}_W$ is obtained. In fact, it is a one-step reduction. We note that the step that estimates the conditional probability can be abstracted and summarized as a separate protocol, and it has an inductive nature. Instead of using it only once, we iteratively use the protocol, which is inspired by the classic divide-and-conquer technique. Thus our successive refinement protocol relaxes the restriction in [14] and achieve an upper bound for a wider parametric range.

The squared loss is the most important loss, in both theoretical analysis and algorithm research. By directly specializing Theorem 1, we have the following upper bounds under the squared loss.

**Corollary 1.** *For the squared loss, we have*

$$R(m,n,k,l,p=2) \preceq \begin{cases} \dfrac{k}{mnl} \vee \dfrac{1}{mn}, & n \geq k, m(l \wedge k) > 1000k \log(mn) \log n, \\[3mm] \dfrac{\log(\frac{k}{n}+1)}{ml} \vee \dfrac{1}{mn}, & \dfrac{k}{2^l} \leq n < k, m(l \wedge n) > 2000n \log(mn) \log n, \\[3mm] \dfrac{k}{mn2^l}, & n < \dfrac{k}{2^l}, m(l \wedge n) > 4000n \log(mn) \log n, \\[3mm] \dfrac{\log k}{ml}, & \log k < l < n, ml < k. \end{cases}$$

Lemma 1 can be specialized to obtain the lower bounds as well. Then we have a more complete characterization of the order of $R(m,n,k,l,p=2)$.

$$R(m,n,k,l,p=2) \asymp \begin{cases} \dfrac{k}{mnl} \vee \dfrac{1}{mn}, & n \geq k, ml \succeq k, \\[3mm] \dfrac{1}{ml} \vee \dfrac{1}{mn}, & \dfrac{k}{2^l} \leq n < k \text{ or } n \geq k, ml \preceq k, \\[3mm] \dfrac{k}{mn2^l}, & n < \dfrac{k}{2^l}, mn2^l \succeq k^2. \end{cases} \tag{4}$$

# 4 Optimal Rates for $p > 2$

For $p > 2$, we first present the upper bound in the following.

**Theorem 3.** *Let $p > 2$, then we have* $R(m, n, k, l, p) \preceq$

$$
\begin{cases}
\dfrac{k}{(mnl)^{\frac{p}{2}}} \vee \dfrac{1}{(mn)^{\frac{p}{2}}}, & n \geq k,\, m(l \wedge k^{\frac{2}{p}}) > 1000k \log(mn) \log n, & \text{(5a)} \\[3ex]
\dfrac{\log^{\frac{p}{2}} k}{(ml)^{\frac{p}{2}} n^{\frac{p}{2}-1}} \vee \dfrac{1}{(mn)^{\frac{p}{2}}}, & \dfrac{k}{(2^l)^{\frac{p}{2}}} \leq n < k,\, m(l \wedge n^{\frac{2}{p}}) \geq 1000n \log(mn) \log k, & \\[1ex]
& \qquad\qquad\qquad\qquad l > \log k, & \text{(5b)} \\[3ex]
\left( \dfrac{k}{mn2^l} \right)^{\frac{p}{2}}, & n < \dfrac{k}{(2^l)^{\frac{p}{2}}},\, m(l \wedge n) > 4000n \log(mn) \log n, & \text{(5c)} \\[3ex]
\dfrac{\log^p k \vee \log^{4p}(mn)}{(ml)^{p-1}} \vee \dfrac{1}{(mn)^{\frac{p}{2}}}, & \log k < l < n,\, ml < n. & \text{(5d)}
\end{cases}
$$

*Proof.* The case (5a) is by Proposition 1 in Appendix A, the case (5b) is by Proposition 4 in Appendix D, the case (5c) is by Proposition 2 in Appendix B, and the case (5d) is by Proposition 5 in Appendix E. We present a sketch of the proof for these cases here.

**Refinement protocol** For the case (5a), the first step of the protocol for $p > 2$ is the same as that for $1 \leq p \leq 2$. That is, a rough estimate $\hat{\boldsymbol{p}}_W$ is established by assigning the first half of all encoders uniformly to estimating each entry $p_W(w)$ using the one-bit protocol in [14]. But it is not enough, since for $p > 2$ the estimation error for the big entry $p_W(w)$ decays significantly slower than that for the small entry, which is different from the case $1 \leq p \leq 2$. To overcome the difficulty, a refinement method is necessary, where a portion of roughly $\hat{p}_W(w)$ remaining budget is allocated to estimate $p_W(w)$. The spirit of the allocation strategy is similar to that designed for the pointwise estimation problem [11] with $n = 1$. Details can be found in Appendix A.

**Refinement protocol with sample compression techniques** For the case (5b), sample compression techniques are further incorporated. The starting point is also the refinement method, but in this case the length of the distribution is too long, namely $k > n$. Hence the optimal estimation method for the encoder is not to summarize its samples and describe each $p_W(w)$, but to describe samples it observes directly. This makes how to do the refinement step obscure.

Sample compression techniques are designed to customize the refinement methods in this regime. Note that the number of the elements $w$ with $p_W(w) \succeq \frac{1}{n}$ (denote the set containing those elements $w$ by $\mathcal{W}'$) is about $n$. Samples are first compressed by projecting them to $\mathcal{W}'$, which saves the communication budget for describing them. Hence those $p_W(w) \succeq \frac{1}{n}$ are refined by invoking the protocol for the case (5a). See Appendix D for details.

**The remaining two cases** The bound in (5c) is a corollary of the successive refinement protocol in Appendix B. For the case (5d), the bound is achieved by a refinement protocol exploiting both sample compression and thresholding techniques, and details can be found in Appendix E.2. $\qquad\square$

Similar to Section 3, we present the lower bound as a baseline in the following lemma.

**Lemma 2.** *For $p > 2$, we have*

$$
R(m, n, k, l, p) \succeq
\begin{cases}
\dfrac{k}{(mnl)^{\frac{p}{2}}} \vee \dfrac{1}{(mn)^{\frac{p}{2}}}, & n \geq k \log k,\, m > \left( \dfrac{k}{l} \right)^2 \\[3ex]
\dfrac{1}{(ml)^{\frac{p}{2}} n^{\frac{p}{2}-1} \log n} \vee \dfrac{1}{(mn)^{\frac{p}{2}}}, & \dfrac{k}{(2^l)^{\frac{p}{2}}} \leq n < k \log k,\, m > \left( \dfrac{n/\log n}{l} \right)^2, \\[3ex]
\dfrac{k}{(mn2^l)^{\frac{p}{2}}}, & n < \dfrac{k}{(2^l)^{\frac{p}{2}}},\, mn2^l > k^2, \\[3ex]
\dfrac{1}{(ml)^{p-1}} \vee \dfrac{1}{(mn)^{\frac{p}{2}}}, & ml < \dfrac{k}{2}.
\end{cases}
$$

*Proof.* Most of the lower bounds can be derived from that under the $\ell^1$ loss in [14] using Hölder's Inequality. Two of the bounds, namely $\frac{1}{(ml)^{\frac{p}{2}} n^{\frac{p}{2}-1} \log n}$ in the second case and $\frac{1}{(ml)^{p-1}}$ in the last case require the additional algebraic technique in the proof of Lemma 1. The last exception, the centralized bound $\frac{1}{(mn)^{\frac{p}{2}}}$ without communication constraints is little-known but easy to show. Moreover, we think its proof uncovers the major differences of the estimation problem for $p > 2$ compared with that for $p \leq 2$. Hence it is sketched as follows. Detailed proof for all the lower bounds can be found in Appendix G.

**The centralized bound without communication constraints**  For $p > 2$, the key observation is that distributions most difficult to estimate have only a few large entries of constant order. It is in contrast to the case $1 \leq p \leq 2$ where such distributions are close to uniform and each entry is roughly $\sim \frac{1}{k}$. In light of this, the bound can be proved by a simple way of reduction to a binary hypothesis testing. It is elaborated in the proof of Lemma 11 in Appendix G. □

*Remark* 4.  We summarize our technical contributions in the lower bounds in Lemmas 1 and 2 here. From a technical perspective, the overall proof of the lower bounds depend on four different ways of reduction to hypothesis testing problems. Most of lower bounds under the $\ell^p$ loss are derived from that under the $\ell^1$ loss in [14]. Typically, the proof in [14] uses the reduction to a hypothesis testing problem of roughly $2^{\frac{k}{2}}$ hypotheses. However, the derived bounds are not tight, especially for the case $p > 2$. In this work, one of the major finding is that the optimal bounds are different for $p \leq 2$ and $p > 2$. To show that, we introduce two major techniques, which rely on three ways of reduction to hypothesis testing. First, the centralized bound $\frac{1}{(mn)^{\frac{p}{2}}}$ for $p > 2$ is proved by the reduction to a binary hypothesis testing. Second, the algebraic technique is exploited for the communicate-constrained bounds $\frac{1}{(ml)^{\frac{p}{2}} n^{\frac{p}{2}-1} \log n}$ and $\frac{1}{(ml)^{p-1}}$. In its spirit, the technique used in these two cases is equivalent to two different ways of reduction hypothesis testing problems, with roughly $2^n$ and $2^{ml}$ hypotheses respectively. The latter three ways of reduction used in this work improve bounds derived from the first way in [14], so that the overall lower bound is tight. These four reductions together complete the proof of lower bounds.

Combining Theorem 3 and lemma 2, the optimal rates can be characterized in the following, except for the third case where our lower and upper bounds do not coincide. We conjecture that the lower bound $\frac{k}{(mn2^l)^{\frac{p}{2}}}$ is tight, which is partially verified for the case $n = 1$ in the next section.

$$
R(m,n,k,l,p) \asymp
\begin{cases}
\dfrac{k}{(mnl)^{\frac{p}{2}}} \vee \dfrac{1}{(mn)^{\frac{p}{2}}}, & n \geq k,\ ml \succeq n \\[2ex]
\dfrac{1}{(ml)^{\frac{p}{2}} n^{\frac{p}{2}-1}} \vee \dfrac{1}{(mn)^{\frac{p}{2}}}, & \dfrac{k}{(2^l)^{\frac{p}{2}}} \leq n < k,\ ml \succeq n, \\[2ex]
\dfrac{k}{(mn2^l)^{\frac{p}{2}}}, & n < \dfrac{k}{(2^l)^{\frac{p}{2}}},\ mn2^l \succeq k^2, \\[2ex]
\dfrac{1}{(ml)^{p-1}} \vee \dfrac{1}{(mn)^{\frac{p}{2}}}, & ml \preceq k, k < n \text{ or } ml \preceq n, k > n.
\end{cases}
\tag{6}
$$

## 5  Optimal Rates for $n = 1$, $p \geq 2$ and the Non-interactive Estimation Protocol

For $n = 1$ and $p \geq 2$, the lower bound can be derived by specializing Lemma 2, and the compatible upper bound is achieved by a non-interactive protocol, shown in the following thoerem.

**Theorem 4.** *Let $n = 1$, $p \geq 2$ and $m(2^l \wedge k^{\frac{2}{p}}) \geq k^2$. We can design a non-interactive protocol that achieves the optimal rate $R(m,1,k,l,p) \asymp \frac{k}{(m2^l)^{\frac{p}{2}}} \vee \frac{1}{m^{\frac{p}{2}}}$.*

*Proof.* The lower bound is implied by Lemma 2. The upper bound is by Proposition 6 in Appendix F, for which we present the proof sketch here.

**Non-interactive protocol with random hashing**   For each encoder, a hash function $h_i : \mathcal{W} \to [1 : 2^l]$ is randomly generated. Then the encoder can compress its sample $W_i$ to the message $h_i(W_i)$ using its $l$ bits. Upon receiving all the messages, the decoder can directly obtain the estimate by constructing and rescaling the histogram. Further discussions can be found in the proof of Proposition 6 and Appendix F.1.                                                                      $\square$

*Remark* 5. Note that the centralized bound $\frac{1}{m^{\frac{p}{2}}}$ without the communication constraints is neglected by previous works [16, 17] (see Theorem 6 in [16] and Corollary 3.2 in [17]). Hence the lower bounds in both works are clearly not tight (for $p > 2$). The work [17] further claimed that the lower bound $\frac{k}{(m2^l)^{\frac{p}{2}}} \vee \frac{k^{1-\frac{p}{2}}}{m^{\frac{p}{2}}}$ is optimal (see Lemma 3.3 therein), but the sketch given there is not sufficient to describe a protocol that achieves the bound. In fact, given that the lower bound in [17] can be strictly improved, it is impossible to show its optimality. Moreover, constructing the protocol that achieves the optimal rates for $p > 2$ is not straightforward and needs additional ideas. We use random hashing technique to resolve the difficulty in this work, and there may be other solutions.

*Remark* 6. We give some intuitive explanations about why our random hashing protocol achieves the optimal rate in Theorem 4, while existing methods like the simulate-and-infer protocol [7, 8, 18] fail to do so. As discussed in Section 1 and the proof sketch of Lemma 2, for $p > 2$ relatively larger entries $p_W(w)$ are typically more difficult to estimate, and communication budgets should be invested more into estimating them. In this sense, the problem resembles a sparse distribution estimation. The simulate-and-infer protocol uses too much communication budget to estimate the smaller entries, while fails to simulate enough samples for estimating the larger entries. In contrast, random hashing reduces estimation errors for the larger entries, despite increasing the error for the smaller entries. Therefore, it achieves an optimal communication budget allocation strategy, as well as the optimal rate.

# 6   Summary of the Optimal Rates

In Table 1, we summarize the characterizations of the optimal rate obtained in Equations (2) to (4) and (6) and Theorem 4, where fundamentally different regimes lead to different rates. The essential bounds originally proved in this work are highlighted in red, while those established in previous works [7, 8, 14, 16, 17] are shown in blue. All the other bounds are corollaries of them. The optimal rates (up to logarithmic factors) are obtained for most cases, except the case $p > 2$, $n < \frac{k}{(2^l)^{\frac{p}{2}}}$ and $mn2^l \geq k^2$, where our lower and upper bounds do not coincide. Though a good news is that for its special case $n = 1$, the optimal rates can be obtained. We conjecture that the lower bound $\frac{k}{(mn2^l)^{\frac{p}{2}}}$ is tight, which is partially verified in the case $n = 1$. We find several interesting phenomena of the optimal rates.

1. There is an elbow effect in the parameter $p$ between the regimes $1 \leq p < 2$ and $p \geq 2$. The difference is clearly reflected in the centralized bound without any communication constraints, i.e. $l = \infty$. The bound is $\frac{k^{1-\frac{p}{2}}}{(mn)^{\frac{p}{2}}}$ for $1 \leq p < 2$, while for $p \geq 2$ it is $\frac{1}{(mn)^{\frac{p}{2}}}$ and independent of the dimension $k$ of the distribution. The other sharp difference is that, for a medium $n$, i.e. $\frac{k}{(2^l)^{\frac{p}{2} \vee 1}} \leq n < k$, the optimal rate is independent of $k$ (up to logarithmic factors) for $p \geq 2$, which is not the case for $1 \leq p < 2$.

2. Second, the minimum transmitted bits required for recovering the same rates in the centralized case without any communication constraints are interesting for $p > 2$. It is roughly $k^{\frac{2}{p}}$ for $k < n$, $ml \geq k$ and $n^{\frac{2}{p}}$ for $k \geq n$, $ml \geq n$, which is out of expectation. It shows a shrinkage compared to the required number of bits $k$ and $n$ for the case $1 \leq p < 2$. Similarly, for $n = 1$ and $m2^l \geq k^2$, the required number of bits is roughly $\frac{2}{p} \log k$ instead of $\log k$.

3. The last observation is that if the total communication budget is extremely tight ($ml \ll k$), then the optimal rate is dependent only on the total budget and independent of the parameters $k$ and $n$. This parameter regime has not been studied in previous work to our best knowledge.

Table 1: Bounds of $R(m, n, k, l, p)$ for Different Cases

| Parameter Regimes | $p = 1$ | $1 \le p \le 2$ | $p = 2$ | $p \ge 2$ |
|---|---|---|---|---|
| $l = \infty$ | $R \asymp \sqrt{\frac{k}{mn}}$ | $R \asymp \frac{k^{1-\frac{p}{2}}}{(mn)^{\frac{p}{2}}}$ | $R \asymp \frac{1}{mn}$ | $R \asymp \frac{1}{(mn)^{\frac{p}{2}}}$ 
 (Lemma 11) |
| $n \ge k$, 
 $l^{\frac{p}{2} \vee 1} \le k$, 
 $ml \ge k$ | $R \asymp \frac{k}{\sqrt{mnl}}$ | $R \asymp \frac{k}{(mnl)^{\frac{p}{2}}}$ | $R \asymp \frac{k}{mnl}$ | $R \asymp \frac{k}{(mnl)^{\frac{p}{2}}}$ 
 (Proposition 1) |
| $\frac{k}{(2^l)^{\frac{p}{2} \vee 1}} \le n < k$, 
 $l^{\frac{p}{2} \vee 1} \le n$, 
 $ml \ge k \ (p \le 2)$, 
 $ml \ge n \ (p > 2)$ | $R \asymp \sqrt{\frac{k}{ml}}$ | $R \asymp \frac{k^{1-\frac{p}{2}}}{(ml)^{\frac{p}{2}}}$ | $R \asymp \frac{1}{ml}$ | $R \asymp \frac{1}{(ml)^{\frac{p}{2}} n^{\frac{p}{2}-1}}$ 
 (Propositions 2 and 4) |
| $ml < k \ (p \le 2$ 
 or $p > 2, k \le n)$, 
 $ml < n \ (p > 2, k > n)$, 
 $l > \log k$ | $R \asymp 1$ | $R \asymp \frac{1}{(ml)^{p-1}}$ 
 (Proposition 5) | $R \asymp \frac{1}{ml}$ | $R \asymp \frac{1}{(ml)^{p-1}}$ 
 (Proposition 5) |
| $n < \frac{k}{(2^l)^{\frac{p}{2} \vee 1}}$, 
 $mn2^l \ge k^2$ | $R \asymp \frac{k}{\sqrt{mn2^l}}$ | $R \asymp \frac{k}{(mn2^l)^{\frac{p}{2}}}$ | $R \asymp \frac{k}{mn2^l}$ | $R \preceq \left(\frac{k}{mn2^l}\right)^{\frac{p}{2}}$ 
 (Proposition 2) 
 $R \succeq \frac{k}{(mn2^l)^{\frac{p}{2}}}$ |
| $n = 1$, 
 $(2^l)^{\frac{p}{2} \vee 1} < k$, 
 $m2^l \ge k^2$ | $R \asymp \frac{k}{\sqrt{m2^l}}$ | $R \asymp \frac{k}{(m2^l)^{\frac{p}{2}}}$ | $R \asymp \frac{k}{m2^l}$ | $R \asymp \frac{k}{(m2^l)^{\frac{p}{2}}}$ 
 (Proposition 6) |

## 7   Discussions and Future Works

In this work, we focused on the minimax optimal rates of distribution estimation over the whole probability simplex, without imposing any additional assumptions on the structure of the distribution to be estimated. In contrast, many previous works studied the structured distribution estimation problems [4, 9–11, 17], such as the point-wise distribution estimation problem [11, 17] and the sparse distribution estimation problems [9, 10]. These problems are also of both theoretical and practical importance. However, existing works limited their scope to $n = 1$, leaving problems with $n > 1$ and $\ell^p$ losses unexplored. We hope our methods can help with determining optimal rates for these problems.

Moreover, the methods in this work are not restricted to the discrete distribution estimation problem. The analysis of statistical learning problems in various other settings under $\ell^p$ losses can also benefit from our methods. The methods deal with the difficulty induced by the normalization constraint of the distribution in the distribution estimation setting, which also shows a potential direction for solving problems with similar implicit constraints. A more challenging problem is whether we can construct non-interactive protocols, instead of interactive protocols in this work, to achieve the minimax optimal rates with $n > 1$ samples per terminal and under $\ell^p$ losses. Determining the privacy-constrained optimal rates for $n > 1$ and $\ell^p$ losses is also an interesting direction for future work.

Finally, our protocol for estimating a discrete distribution (especially for the squared loss) can be used as a subroutine of the protocol achieving the optimal rates in the nonparametric density estimation and regression problems. See the works [20, 21] for details.

## Acknowledgments and Disclosure of Funding

This work was supported in part by the NSFC Projects No.12025104 and 62301144, in part by the SEU Startup Fund No.RF1028623030, and in part by the Zhishan Young Scholar Fund No.2242025RCB0032.

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

# Appendix

This technical appendix is devoted to presenting the detailed proof of the main results, by designing optimal protocols to achieve the upper bounds for different parameter regimes in Appendices A to F and deriving the compatible (up to logarithmic factors) lower bounds in Appendix G. These sections are organized as in Table 1 and follows.

- Appendix A presents the refinement protocol for cases (1a) and (5a) in Theorems 1 and 3, summarized in Proposition 1.
- Appendix B presents the successive refinement protocol with adaptive budget allocation for cases (1b) and (1c) in Theorem 1 and (5c) in Theorem 3, summarized in Proposition 2.
- Appendix C presents the non-interactive successive refinement protocol for the TV loss in Theorem 2, summarized in Proposition 3.
- Appendix D presents the refinement protocol with sample compression techniques for the case (5b) in Theorem 3, summarized in Proposition 4.
- Appendix E presents the refinement protocol with thresholding for cases (1d) and (5d) in Theorems 1 and 3, summarized in Proposition 5.
- Appendix F presents the non-interactive protocol based on random hashing for the $n = 1$ case in Theorem 4, summarized in Proposition 6.
- Appendix G shows all the lower bounds in Lemmas 1 and 2.

## A  The Protocol for Cases (1a) and (5a) and Its Analysis

In this section, we design the estimation protocol with refinement methods that achieves the optimal rates for cases (1a) and (5a), summarized in the following proposition.

**Proposition 1.** *Let $p \geq 2$, $k \leq n$, $ml > 1000k \log(mn) \log n$ and $l \leq k^{\frac{2}{p}}$. Then for the estimation problem in Section 2, there exists an interactive refinement protocol $\mathrm{IR}(m, n, k, l, p)$ such that for any $\boldsymbol{p}_W \in \Delta_{\mathcal{W}}$, the protocol outputs an estimate $\hat{\boldsymbol{p}}_W$ satisfying $\mathbb{E}[\|\hat{\boldsymbol{p}}_W - \boldsymbol{p}_W\|_p^p] = O\left(\frac{k}{(mnl)^{\frac{p}{2}}}\right)$.*

*Remark* 7. With the help of Proposition 1, for $1 \leq p < 2$, let the protocol $\mathrm{IR}(m, n, k, l, p)$ be the same as that for $p = 2$, i.e., $\mathrm{IR}(m, n, k, l, 2)$. Then by the Hölder's inequality, we have

$$\mathbb{E}[\|\hat{\boldsymbol{p}}_W - \boldsymbol{p}_W\|_p^p] \leq k^{1-\frac{p}{2}} \left(\mathbb{E}[\|\hat{\boldsymbol{p}}_W - \boldsymbol{p}_W\|_2^2]\right)^{\frac{p}{2}}.$$

Hence

$$R(m, n, k, l, p) \leq k^{1-\frac{p}{2}} R(m, n, k, l, 2)^{\frac{p}{2}},$$

and the minimax upper bound for $1 \leq p < 2$ is easily implied by that for $p = 2$.

Now we return to the proof of Proposition 1. Each entry of the distribution can be estimated by invoking the one-bit protocol in [14] for the estimation of a binary distribution. We first show the error bound in the following lemma, which can be proved by adapting the proof of Theorem A.2 and A.3 therein.

**Lemma 3.** *Suppose that there are $m'$ users and each of them observe an i.i.d. sample from the binary distribution $\mathrm{B}(n, q)$ and $m' > 1000 \log n$. Then for $p \geq 2$, there exists a one-bit protocol which outputs an estimate $\hat{q}$ satisfying*

$$\mathbb{E}\left[|q - \hat{q}|^p\right] = O\left(\left(\frac{q}{m'n}\right)^{\frac{p}{2}} + \frac{q}{(m'n)^{p-1}} + \left(\frac{q}{n} \vee \frac{1}{n^2}\right)^{\frac{p}{2}} e^{-\frac{m'}{240 \log n}}\right). \tag{7}$$

### A.1  The Refinement Protocol

**Rough estimation**   The first step is to let the first $\frac{m}{2}$ encoders and the decoder jointly generate a rough estimate $\hat{\boldsymbol{p}}_W^1$. Let $m' = \lfloor \frac{ml}{2k} \rfloor$. Each encoder can concurrently run $l$ one-bit protocols in Lemma 3 using its $l$ bits, where $l \leq k^{\frac{2}{p}} \leq k \leq n$ and the goal of each protocol is to estimate $p_W(w)$ for some $w \in \mathcal{W}$. At the same time, a proper allocation plan can ensure that for each $w \in \mathcal{W}$, there are $m'$ encoders running the protocol for estimating $p_W(w)$. The decoder then obtains the rough estimate $\hat{\boldsymbol{p}}_W^1$.

**Refinement of the estimate** The second step is to let the next $\frac{m}{2}$ encoders and the decoder jointly generate a refined estimate $\hat{p}_W^2$. Let $m(w) = \lfloor \frac{ml(\hat{p}_W^1(w) + \frac{1}{k})}{4} \rfloor \wedge \frac{m}{2}$. Each encoder can concurrently run $l$ one-bit protocols in Lemma 3 using its $l$ bits, for estimating some $p_W(w)$. At the same time, a proper allocation plan can ensure that for each $w \in \mathcal{W}$, there are $m(w)$ encoders[1] running the protocol for estimating $p_W(w)$. The decoder then constructs the refined estimate $\hat{p}_W^2$ following the protocol.

*Remark* 8 (Necessity of the Refinement Methods). It is easy to analyze the error of the rough estimate $\hat{p}_W^1$. By Lemma 3 and the assumption $ml > 1000k \log(mn) \log n$, for any $w \in \mathcal{W}$ we have

$$
\mathbb{E}\left[|p_W(w) - \hat{p}_W^1(w)|^p\right] = O\left(\left(\frac{kp_W(w)}{mnl}\right)^{\frac{p}{2}} + \left(\frac{k}{mnl}\right)^{p-1} p_W(w) + \left(\frac{1}{mnl}\right)^{\frac{p}{2}}\right). \quad (8)
$$

However, simply taking the summation can only get the total error bound $O((\frac{k}{mnl})^{\frac{p}{2}})$, which is not tight for $p > 2$. To obtain the tight bound, our protocol uses the rough estimate $\hat{p}_W^1$ for directing the budget allocation in the second step. Then the refined estimate in the second step can achieve the desired upper bound, i.e. $\mathbb{E}[\|\hat{p}_W^2 - p_W\|_p^p] = O\left(\frac{k}{(mnl)^{\frac{p}{2}}}\right)$, which completes the proof of Proposition 1. See Appendix A.2 for details.

## A.2 Proof of Proposition 1: Error Analysis for the Protocol in Appendix A.1

We first show the following preliminary error bound concerning the rough estimate.

**Lemma 4.** If $p_W(w) \geq \frac{1}{k}$ for some $w \in \mathcal{W}$, then $\mathbb{P}\left[\frac{p_W(w)}{\hat{p}_W^1(w)} \geq 2\right] \leq O\left(\frac{1}{(np_W(w))^{\frac{p}{2}}}\right)$.

*Proof.* By (8) and $p_W(w) \geq \frac{1}{k}$, we have

$$
\mathbb{E}\left[|p_W(w) - \hat{p}_W^1(w)|^p\right] = O\left(\left(\frac{kp_W(w)}{mnl}\right)^{\frac{p}{2}}\right). \quad (9)
$$

By the Markov inequality, we can obtain that

$$
\mathbb{P}\left[\frac{p_W(w)}{\hat{p}_W^1(w)} \geq 2\right] = \mathbb{P}\left[\frac{\hat{p}_W^1(w)}{p_W(w)} \leq \frac{1}{2}\right] \leq \mathbb{P}\left[|\hat{p}_W^1(w) - p_W(w)| \geq \frac{1}{2}p_W(w)\right]
$$
$$
\leq \frac{2^p \mathbb{E}[|\hat{p}_W^1(w) - p_W(w)|^p]}{p_W(w)^p}.
$$

Then by (9) and the assumption that $ml > 1000k \log(mn) \log n$, we complete the proof. □

Now we return to the proof of Proposition 1. Note that it suffices to show that for each $w \in \mathcal{W}$,

$$
\mathbb{E}\left[|p_W(w) - \hat{p}_W^2(w)|^p\right] = O\left(\frac{1}{(mnl)^{\frac{p}{2}}} + \left(\frac{k}{mnl}\right)^{p-1} p_W(w) + \frac{p_W(w)^{\frac{p}{2}}}{(mn)^{\frac{p}{2}}}\right), \quad (10)
$$

then taking the summation and using $mnl \geq k^2$ can complete the proof.

---

[1] One may worry that the estimate $\hat{p}_W^1$ may not be normalized. But it does not affect the subsequent steps of using $\hat{p}_W^1$ for directing the budget allocation. This can be seen by the following analysis. By the proof of Theorem A.2 in [14] and $n \geq k$, for a constant $C > 1$, $\mathbb{P}[\|\hat{p}_W^1\|_1 \geq C] \leq \sum_w \mathbb{P}[|\hat{p}_W^1(w) - p_W(w)| \geq (C-1)(\frac{1}{n} \vee \sqrt{\frac{p_W(w)}{n}})] \leq k \log n \cdot e^{-\frac{m'}{240 \log n}}$, which is sufficiently small if $ml \gg k \log n \log(mn)$. In the case that $\hat{p}_W^1$ is used as a ratio for budget allocation, we can simply divide it by the constant $C$ and then the error analysis is still true. Hence, for simplicity we assume that $\hat{p}_W^1$ is normalized and do not point out this minor obstacle in similar cases where $\hat{p}_W^1$ is generated by the protocol in Lemma 3.

By Lemma 3, we have

$$\mathbb{E}\left[|p_W(w) - \hat{p}_W^2(w)|^p\right]$$
$$= O\left(\mathbb{E}\left[\left(\frac{p_W(w)}{mnl(\hat{p}_W^1(w) + \frac{1}{k})}\right)^{\frac{p}{2}}\right] + \left(\frac{k}{mnl}\right)^{p-1} p_W(w) + \left(\frac{1}{mnl}\right)^{\frac{p}{2}} + \left(\frac{p_W(w)}{mn}\right)^{\frac{p}{2}}\right).$$

It suffices to bound the first term. Define the event $\mathcal{F}_w = \left\{\frac{p_W(w)}{\hat{p}_W^1(w)} \geq 2\right\}$. Then by Lemma 4 and $n \geq k$, we have

$$\mathbb{E}\left[\left(\frac{p_W(w)}{mnl(\hat{p}_W^1(w) + \frac{1}{k})}\right)^{\frac{p}{2}}\right]$$
$$= \mathbb{E}\left[\mathbb{1}_{\mathcal{F}_w}\left(\frac{p_W(w)}{mnl(\hat{p}_W^1(w) + \frac{1}{k})}\right)^{\frac{p}{2}}\right] + \mathbb{E}\left[\mathbb{1}_{\mathcal{F}_w^{\complement}}\left(\frac{p_W(w)}{mnl(\hat{p}_W^1(w) + \frac{1}{k})}\right)^{\frac{p}{2}}\right]$$
$$\leq \mathbb{P}\left[\mathcal{F}_w\right] \cdot \left(\frac{kp_W(w)}{mnl}\right)^{\frac{p}{2}} + O\left(\left(\frac{1}{mnl}\right)^{\frac{p}{2}}\right)$$
$$= \mathbb{1}_{\{p_W(w) < \frac{1}{k}\}} \cdot O\left(\left(\frac{1}{mnl}\right)^{\frac{p}{2}}\right) + \mathbb{1}_{\{p_W(w) \geq \frac{1}{k}\}} \cdot O\left(\left(\frac{1}{np_W(w)} \cdot \frac{kp_W(w)}{mnl}\right)^{\frac{p}{2}}\right) + O\left(\left(\frac{1}{mnl}\right)^{\frac{p}{2}}\right)$$
$$= O\left(\left(\frac{1}{mnl}\right)^{\frac{p}{2}}\right),$$

which completes the proof.

## B  The Protocol for Cases (1b), (1c) and (5c) and Its Analysis

In this section, we design a successive refinement protocol with adaptive budget allocation that achieves the optimal rates for cases (1b), (1c) and (5c). Similar to the discussion in Remark 7, it suffices to show the following proposition for $p \geq 2$.

**Proposition 2.** *Let $p \geq 2$. Then for the problem in Section 2, there exists an interactive protocol* $\mathrm{SSR}(m,n,k,l,p)$ *such that for any $\boldsymbol{p}_W \in \Delta_{\mathcal{W}}$, the protocol outputs an estimate $\hat{\boldsymbol{p}}_W$ satisfying,*

1. *if $k \leq n$, $m(l \wedge k) > 1000k\log(mn)\log n$, then $\mathbb{E}[\|\hat{\boldsymbol{p}}_W - \boldsymbol{p}_W\|_p^p] = O\left(\left(\frac{k}{mnl}\right)^{\frac{p}{2}} \vee \frac{1}{(mn)^{\frac{p}{2}}}\right)$;*

2. *if $n < k \leq (2^l - 1) \cdot n$, $l \geq 2$ and $m(l \wedge n) > 2000n\log(mn)\log n$, then $\mathbb{E}[\|\hat{\boldsymbol{p}}_W - \boldsymbol{p}_W\|_p^p] = O\left(\left(\frac{\log(\frac{k}{n}+1)}{ml}\right)^{\frac{p}{2}} \vee \frac{1}{(mn)^{\frac{p}{2}}}\right)$;*

3. *if $k > (2^l - 1) \cdot n$, $l \geq 4$ and $m(l \wedge n) > 4000n\log(mn)\log n$, then $\mathbb{E}[\|\hat{\boldsymbol{p}}_W - \boldsymbol{p}_W\|_p^p] = O\left(\left(\frac{k}{2^l mn}\right)^{\frac{p}{2}}\right)$.*

*Remark* 9. Although the bound in Proposition 2 is not always tight for $p > 2$, it is indeed tight (up to logarithmic factors) for $p = 2$ and can imply tight bound for $1 \leq p < 2$. The advantage of using the successive refinement protocol for $1 \leq p < 2$ is that the protocol can apply for a lager parameter regime. In comparison, the protocol in Appendix D can be used for $1 \leq p < 2$ and $k > n$ but it requires that $l > \log k$. Hence it fails to handle the case 2 for $\log(\frac{k}{n} + 1) < l \leq \log k$ and the case 3 in Proposition 2.

We design the successive refinement protocol $\mathrm{SSR}(m,n,k,l,p)$ in Proposition 2 inductively, which turns out to be a successive refinement procedure. The protocol for each case in Proposition 2 relies on that for the preceding case. The goal is to estimate a distribution $\boldsymbol{p}_W \in \Delta_{\mathcal{W}}$. If the communication budget $l$ for each encoder is too tight, then it is difficult to describe all the entries of $\boldsymbol{p}_W$. Instead, we can perform a a divide-and-conquer technique.

At each step, choose some $l_0$ and construct a division $\mathcal{W} = \cup_{s=1}^{t}\mathcal{W}_s$ with $|\mathcal{W}_s| \leq 2^{l_0} - 1$, $l_0 \leq l$ and $t = \lceil \frac{k}{2^{l_0}-1} \rceil$. Then each encoder is assigned several $\mathcal{W}_s$ and ordered to describe the conditional distribution $\boldsymbol{p}_s \in \Delta_{\mathcal{W}_s}$ for the assigned $\mathcal{W}_s$, where $p_s(w) \triangleq p(w|\mathcal{W}_s)$. Based on the message, the decoder constructs $\hat{\boldsymbol{p}}_s$ as an estimate of $\boldsymbol{p}_s$. Let the block distribution be $\boldsymbol{p}_B$, where $p_B(s) = \sum_{w \in \mathcal{W}_s} p(w)$. As long as an estimate $\hat{\boldsymbol{p}}_B$ of the distribution $\boldsymbol{p}_B$ can be obtained, we can obtain an estimate $p_W(w) = \hat{p}_B(s)\hat{p}_s(w)$ for $w \in \mathcal{W}_s$.

The above procedure can be repeated for the estimation of $\boldsymbol{p}_B$. Note that $\boldsymbol{p}_B \in \Delta_{[1:t]}$ always has a lower dimension $t$ than the dimension $k$ for $p_W$, the inductive procedure will finally terminate. Hence the estimate $\hat{\boldsymbol{p}}_B$ can be obtained, as well as $\hat{\boldsymbol{p}}_W$.

The error of each one-step procedure is bounded by the following lemma, proved in Appendix B.3.

**Lemma 5.** *For $p \geq 2$, we have*

$$\mathbb{E}[\|\hat{\boldsymbol{p}}_W - \boldsymbol{p}_W\|_p^p] \leq 2^{p-1}\left(\mathbb{E}[\|\hat{\boldsymbol{p}}_B - \boldsymbol{p}_B\|_p^p] + \sum_{s=1}^{t}\mathbb{E}[p_B(s)^{\frac{p}{2}}\hat{p}_B(s)^{\frac{p}{2}}\|\hat{\boldsymbol{p}}_s - \boldsymbol{p}_p\|_p^p]\right). \quad (11)$$

*Remark* 10. For the TV distance ($p = 1$), it is easy to obtain that (cf. Lemma 3.1 in [14])

$$\mathbb{E}[\|\hat{\boldsymbol{p}}_W - \boldsymbol{p}_W\|_{\text{TV}}] \leq \mathbb{E}[\|\hat{\boldsymbol{p}}_B - \boldsymbol{p}_B\|_{\text{TV}}] + \sum_{s=1}^{t}p_B(s)\mathbb{E}[\|\hat{\boldsymbol{p}}_s - \boldsymbol{p}_s\|_{\text{TV}}]. \quad (12)$$

Now consider the subroutine for estimating all the $\boldsymbol{p}_s$, $s = 1, ..., t$ given an estimate $\hat{\boldsymbol{p}}_B$ for $\boldsymbol{p}_B$. By (11), it is intuitive that the budgets for estimating each $\boldsymbol{p}_s$ should be based on the multiplicative weight $\hat{p}_B(s)^{\frac{p}{2}}p_B(s)^{\frac{p}{2}}$ of the estimation error $\|\hat{\boldsymbol{p}}_s - \boldsymbol{p}_s\|_p^p$. It turns out that the number of encoders for estimating $\boldsymbol{p}_s$ can be proportional to $\hat{p}_B(s)$. Since the quantity $\hat{p}_B(s)$ can be obtained by the decoder, the allocation of encoders can be based on it by interaction between the decoder and encoders. Such an allocation plan is in contrast to the estimation problem under the TV loss discussed in Appendix C. The difference is characterized by the error bound (12), where the weight is simply $p_B(s)$ and a uniform budget allocation plan among all the $\boldsymbol{p}_s$, $s = 1, ..., t$ is optimal.

The detailed subroutine is presented in the following subsection.

### B.1 Successive Refinement Subroutines

Suppose that there are $m'$ encoders and each of them observes i.i.d. samples $W^n$. Fix $l_0 \leq l$ and let $n_0 = \lfloor \frac{l}{l_0} \rfloor \wedge n$. Then we design the successive refinement subroutine $\text{SSRSub}(m', n, k, l, l_0, p)$ as follows. It receives an estimate $\hat{\boldsymbol{p}}_B$ of the block distribution $\boldsymbol{p}_B$ of dimension $t$, and outputs an estimate $\hat{\boldsymbol{p}}_W$ of the original distribution $\boldsymbol{p}_W$.

**Allocating frames to blocks**   Divide the $l$-bit message for each encoder into multiple $l_0$-bit frames. Then each encoder holds at least $n_0$ such frames and all encoders hold $m'n_0$ frames in total. Each $l_0$-bit frame is sufficient to transmit a sample, given that the sample is from a fixed block $s$ of size no more than $2^l - 1$. Simply let

$$r(s) = \hat{p}_B(s). \quad (13)$$

Then $\boldsymbol{r}$ is a block distribution. And we allocate all $m'n_0$ frames held by $m'$ encoders to encoding samples in different $\mathcal{W}_s$, such that

(i) for each block $s$, $N_s = \lfloor m'n_0 r(s) \rfloor$ frames are allocated;

(ii) for each encoder, there are at most $\lceil n_0 r(s) \rceil$ frames allocated to transmitting samples in $\mathcal{W}_s$.

**Encoding**   For each block $s$, each encoder divides all its $n$ samples into $\lceil n_0 r(s) \rceil$ parts, and each part has $\lfloor \frac{n}{\lceil n_0 r(s) \rceil} \rfloor$ samples (ignoring the remaining $n - \lceil n_0 r(s) \rceil \cdot \lfloor \frac{n}{\lceil n_0 r(s) \rceil} \rfloor$). Each frame that is held by the encoder and allocated for transmitting samples in block $s$ is then mapped to one of these parts injectively. If in that part, there are samples falling into the block $s$, then the encoder uses the corresponding frame to encode the first such sample. If not, the frame is encoded as 0.

**Decoding and estimating** For each block $s$, the decoder extracts frames in messages which are allocated to the block. For $b = 1, ..., N_s$, let $\tilde{W}_b^s = \emptyset$ if the $b$-th such frame is 0 and let $\tilde{W}_b^s$ be the sample encoded by the frame if it is not 0. The decoder computes $N_s' = \sum_{b=1}^{N_s} \mathbb{1}_{\tilde{W}_b^s \neq \emptyset}$. Then it computes

$$\hat{p}_s(w) = \frac{\sum_{b=1}^{N_s} \mathbb{1}_{\tilde{W}_b^s = w}}{N_s'} \tag{14}$$

if $N_s' \neq 0$, and it computes $\hat{p}_s(w) = \frac{1}{|\mathcal{W}_s|}$ otherwise. Finally, for each $s = 1, ..., t$ and each $w \in \mathcal{W}_s$, it computes $\hat{p}_W(w) = \hat{p}_B(s)\hat{p}_s(w)$.

The complete successive refinement subroutine $\mathrm{SSRSub}(m', n, k, l, l_0, p)$ is summarized in Algorithm 1. The estimation error induced by the subroutine is described in the following lemma, proved in Appendix B.4.

**Lemma 6.** *For $p \geq 2$, we have*

$$\sum_{s=1}^{t} \mathbb{E}[p_B(s)^{\frac{p}{2}} \hat{p}_B(s)^{\frac{p}{2}} \|\hat{\boldsymbol{p}}_s - \boldsymbol{p}_s\|_p^p] = O\left( \frac{\left(1 \vee \frac{t}{n^{\frac{p}{2}}}\right) \cdot \left(\frac{l_0}{l} \vee \frac{1}{n}\right)^{\frac{p}{2}}}{m'^{\frac{p}{2}}} \right). \tag{15}$$

## B.2 Construction of the Complete Protocol SSR

By inductively using the subroutine, the complete protocol $\mathrm{SSR}(m, n, k, l, p)$ for the three cases in Proposition 2 can be constructed as follows. Then the error bounds are derived accordingly from Lemmas 5 and 6 in Appendix B.5 and B.6.

### B.2.1 The Protocol for Case 1

Invoke the first step of the protocol $\mathrm{IR}(m, n, k, l \wedge k, p)$ in Appendix A and then output the rough estimate $\hat{\boldsymbol{p}}_W^1$. By the analysis in Remark 8, we have $\mathbb{E}[\|\hat{\boldsymbol{p}}_W - \boldsymbol{p}_W\|_p^p] = O\left( \left(\frac{k}{mnl}\right)^{\frac{p}{2}} \vee \frac{1}{(mn)^{\frac{p}{2}}} \right)$.

### B.2.2 The Protocol for Case 2

Let $l_0 = \lceil \log(\frac{k}{n} + 1) \rceil \leq l$ and divide the set $\mathcal{W}$ into $t = \lceil \frac{k}{2^{l_0}-1} \rceil \in [\frac{n}{2}, n]$ blocks.

Let the first $\frac{m}{2}$ encoders and the decoder estimate the reduced distribution of dimension $t \leq n$. By the assumptions $m(l \wedge n) > 2000n \log(mn) \log n$, they can invoke the protocol $\mathrm{SSR}(\frac{m}{2}, n, t, l, p)$ in Appendix B.2.1.

Then let the second $\frac{m}{2}$ encoders and the decoder invoke the subroutine $\mathrm{SSRSub}(\frac{m}{2}, n, k, l, l_0, p)$ and compute the estimate of the original distribution $\boldsymbol{p}_W$.

### B.2.3 The Protocol for Case 3

It suffices to design the protocol for $m \geq \frac{8k}{n2^l}$, since the upper bound is vacuous otherwise. Let $l_0 = l$ and then compute the integer $a$ as follows. Let $k_1 = k$, then iteratively compute $k_{u+1} = \lceil \frac{k_u}{2^l-1} \rceil$ for $u = 1, ..., a$. Let $a$ be the minimal number satisfying $k_{a+1} \leq n \cdot (2^l - 1)$, then $k_{a+1} > n$.

Let the first $\frac{m}{2}$ encoders invoke the protocol $\mathrm{SSR}(\frac{m}{2}, n, k, l, p)$ defined in Appendix B.2.2 to estimate the last reduced block distribution of dimension $k_{a+1}$.

Divide the second $\frac{m}{2}$ encoders into $a$ parts, such that the $u$-th part has $m_u = \lfloor \frac{m}{2^{u+1}} \rfloor$ encoders. By the choice of $a$, we have $a \leq \left\lceil \frac{2\log(\frac{k}{n(2^l-1)})}{l} \right\rceil$. Then we have $2^a \leq 2\left(\frac{k}{n(2^l-1)}\right)^{\frac{2}{l}} \leq \frac{m}{2}$ for $l \geq 4$, Hence $m_u \geq \frac{m}{2^{a+1}} \geq 1$. For $u = 1, ..., a$, the decoder iteratively invokes $\mathrm{SSRSub}(m_u, n, k_u, l, l_0, p)$ with encoders in the $u$-th part successively. Then compute the estimate of the original distribution $\boldsymbol{p}_W$.

---
**Algorithm 1** Successive Refinement Subroutine $\mathrm{SSRSub}(m', n, k, l, l_0, p)$
---
**Input:** Parameters $(m', n, k, l, l_0, p)$, an estimate $\hat{\boldsymbol{p}}_B$ of the block distribution $\boldsymbol{p}_B$ (at all encoder and decoder sides).

**Output:** An estimate $\hat{\boldsymbol{p}}_W$ of the original distribution $\boldsymbol{p}_W$.

**Allocating frames to blocks:**

1: $n_0 \leftarrow \lfloor \frac{l}{l_0} \rfloor \wedge n$.

2: Divide each $l$-bit message into $n_0$ frames of length $l_0$.

3: **for** $s = 1 : t$ **do**

4:     $r(s) \leftarrow \hat{p}_B(s)$.

5:     $N_s \leftarrow \lfloor m' n_0 r(s) \rfloor$.

6:     Allocate $N_s$ frames to $\mathcal{W}_s$, s.t. at most $\lceil n_0 r(s) \rceil$ frames are at the same encoder side.

7: **end for**

**Encoding at each encoder side:**

8: **for** $s = 1 : t$ **do**

9:     Divide all $n$ samples into $\lceil n_0 r(s) \rceil$ parts, each with $\lfloor \frac{n}{\lceil n_0 r(s) \rceil} \rfloor$ samples.

10:     Find frames allocated to $\mathcal{W}_s$.

11:     **for** $b = 1 : \lceil n_0 r(s) \rceil$ **do**

12:         **if** all such frames have been encoded **then**

13:             Break.

14:         **else if** $\exists W_i \in \mathcal{W}_s$ for some $W_i$ in the $b$-th part **then**

15:             The $b$-th frame $\leftarrow$ the first such $W_i$.

16:         **else**

17:             The $b$-th frame $\leftarrow 0$.

18:         **end if**

19:     **end for**

20: **end for**

**Decoding and estimating at the decoder side:**

21: **for** $s = 1 : t$ **do**

22:     Extract all $N_s$ frames allocated to $\mathcal{W}_s$.

23:     **for** $b = 1, ..., N_s$ **do**

24:         **if** the $b$-th frame is 0 **then**

25:             $\tilde{W}_b^s \leftarrow \emptyset$.

26:         **else**

27:             $\tilde{W}_b^s \leftarrow$ the $b$-th frame.

28:         **end if**

29:     **end for**

30:     $N_s' \leftarrow \sum_{b=1}^{N_s} \mathbb{1}_{\tilde{W}_b^s \neq \emptyset}$.

31:     **for** $w \in \mathcal{W}_s$ **do**

32:         **if** $N_s' \neq 0$ **then**

33:             $\hat{p}_s(w) \leftarrow \dfrac{\sum_{b=1}^{N_s} \mathbb{1}_{\tilde{W}_b^s = w}}{N_s'}$

34:         **else**

35:             $\hat{p}_s(w) \leftarrow \frac{1}{|\mathcal{W}_s|}$.

36:         **end if**

37:         $\hat{p}_W(w) \leftarrow \hat{p}_B(s)\hat{p}_s(w)$.

38:     **end for**

39: **end for**

40: **return** $\hat{\boldsymbol{p}}_W$.
---

### B.3   Proof of Lemma 5

Note that

$$(p_B(s)p_s(w) - \hat{p}_B(s)\hat{p}_s(w))^2$$
$$\leq (p_B(s)p_s(w) - \hat{p}_B(s)\hat{p}_s(w))^2 + (p_B(s)\hat{p}_s(w) - \hat{p}_B(s)p_s(w))^2$$

$$=(p_s(w)^2 + \hat{p}_s(w)^2)(p_B(s) - \hat{p}_B(s))^2 + 2p_B(s)\hat{p}_B(s)(p_s(w) - \hat{p}_s(w))^2.$$

Then by the Hölder's inequality, we have

$$(p_B(s)p_s(w) - \hat{p}_B(s)\hat{p}_s(w))^p$$
$$\leq 2^{\frac{p}{2}-1}\left[\left(p_s(w)^2 + \hat{p}_s(w)^2\right)^{\frac{p}{2}}(p_B(s) - \hat{p}_B(s))^p + 2^{\frac{p}{2}}p_B(s)^{\frac{p}{2}}\hat{p}_B(s)^{\frac{p}{2}}(p_s(w) - \hat{p}_s(w))^p\right]$$
$$\leq 2^{p-1}\left[\frac{1}{2}\left(p_s(w) + \hat{p}_s(w)\right)(p_B(s) - \hat{p}_B(s))^p + p_B(s)^{\frac{p}{2}}\hat{p}_B(s)^{\frac{p}{2}}(p_s(w) - \hat{p}_s(w))^p\right].$$

where the last inequality is since $p \geq 2$ and $p_s(w), \hat{p}_s(w) \in [0,1]$. Take the summation, and then we have

$$\|\hat{\boldsymbol{p}}_W - \boldsymbol{p}_W\|_p^p \leq 2^{p-1}\sum_{s=1}^t\left[(p_B(s) - \hat{p}_B(s))^p + p_B(s)^{\frac{p}{2}}\hat{p}_B(s)^{\frac{p}{2}}\|\hat{\boldsymbol{p}}_s - \boldsymbol{p}_s\|^p\right].$$

Then (11) is obtained by taking the expectation. We complete the proof.

### B.4   Proof of Lemma 6

If $m'n_0r(s) = m'n_0\hat{p}_B(s) \leq 4$, since $\|\hat{\boldsymbol{p}}_s - \boldsymbol{p}_s\|_p^p \leq 2$, then

$$p_B(s)^{\frac{p}{2}}\hat{p}_B(s)^{\frac{p}{2}}\|\hat{\boldsymbol{p}}_s - \boldsymbol{p}_s\|_p^p \leq 2p_B(s)^{\frac{p}{2}}\hat{p}_B(s)^{\frac{p}{2}} \leq 2^{p+1}\left(\frac{p_B(s)}{m'n_0}\right)^{\frac{p}{2}}.$$

Otherwise, we have $m'n_0r(s) = m'n_0\hat{p}_B(s) > 4$, hence $N_s = \Theta\left(m'n_0r(s)\right) = \Theta\left(m'n_0\hat{p}_B(s)\right)$. Given $\hat{\boldsymbol{p}}_B$, then $\tilde{W}_u^s$ for $u = 1, ..., N_s$ are i.i.d. random variables with

$$q_s \triangleq \mathbb{P}[\tilde{W}_u^s \neq \emptyset | \hat{\boldsymbol{p}}_B] = 1 - (1 - p_B(s))^{\lfloor \frac{n}{\lceil n_0 r(s)\rceil}\rfloor}$$
$$= \Theta\left(p_B(s)\left\lfloor\frac{n}{\lceil n_0 r(s)\rceil}\right\rfloor \wedge 1\right) = \Theta\left(p_B(s)\left\lfloor\frac{n}{\lceil n_0\hat{p}_B(s)\rceil}\right\rfloor \wedge 1\right). \tag{16}$$

In this case, we can establish the bound shown in the following lemma.

**Lemma 7.** $\mathbb{E}[p_B(s)^{\frac{p}{2}}\hat{p}_B(s)^{\frac{p}{2}}\|\hat{\boldsymbol{p}}_s - \boldsymbol{p}_s\|_p^p|\hat{\boldsymbol{p}}_B] \leq C\mathbb{E}\left[\left(\frac{\hat{p}_B(s)}{m'n} \vee \frac{1}{m'nn_0} \vee \frac{p_B(s)}{m'n_0}\right)^{\frac{p}{2}}\Big|\hat{\boldsymbol{p}}_B\right]$ *for some* $C > 0$.

*Proof.* By the Chernoff bound, we have

$$\mathbb{P}\left[N_s' \geq \frac{N_sq_s}{2}\Big|\hat{\boldsymbol{p}}_B\right] \leq \exp\left(-\frac{N_sq_s}{8}\right). \tag{17}$$

And conditional on the event $\{\tilde{W}_u^s \neq \emptyset\}$, the distribution of $\tilde{W}_u^s$ is $\boldsymbol{p}_s$. Hence for each $w \in \mathcal{W}_s$, it is folklore that (cf. Theorem 4 in [22] or Rosenthal's inequality [23]),

$$\mathbb{E}[|\hat{p}_s(w) - p_s(w)|^p|N_s', \hat{\boldsymbol{p}}_B] = O\left(\left(\frac{p_s(w)}{N_s'}\right)^{\frac{p}{2}} + \frac{p_s(w)}{N_s'^{p-1}}\right).$$

Take the summation, since $p \geq 2$ and $p_s(w) \in [0, 1]$ we have

$$\mathbb{E}\left[\|\hat{\boldsymbol{p}}_s - \boldsymbol{p}_s\|_p^p\Big|N_s' \geq \frac{N_sq_s}{2}, \hat{\boldsymbol{p}}_B\right] = O\left(\frac{1}{N_s^{\frac{p}{2}}q_s^{\frac{p}{2}}}\right).$$

Since $\|\hat{\boldsymbol{p}}_s - \boldsymbol{p}_s\|^2 \leq 2$, we have

$$\mathbb{E}[\|\hat{\boldsymbol{p}}_s - \boldsymbol{p}_s\|^2|\hat{\boldsymbol{p}}_B] \leq 2\exp\left(-\frac{N_sq_s}{8}\right) + O\left(\frac{1}{N_s^{\frac{p}{2}}q_s^{\frac{p}{2}}}\right) = O\left(\frac{1}{N_s^{\frac{p}{2}}q_s^{\frac{p}{2}}}\right).$$

Since $n_0 \leq n$, we have $\lceil n_0 \hat{p}_B(s) \rceil \leq n$ and $\frac{n}{\lceil n_0 \hat{p}_B(s) \rceil} \geq 1$. Hence there exists some $C > 0$, such that

$$\mathbb{E}[p_B(s)^{\frac{p}{2}} \hat{p}_B(s)^{\frac{p}{2}} \|\hat{\boldsymbol{p}}_s - \boldsymbol{p}_s\|_p^p | \hat{\boldsymbol{p}}_B]$$

$$\leq C\mathbb{E}\left[\left(\frac{p_B(s)\hat{p}_B(s)}{m' n_0 \hat{p}_B(s) q_s}\right)^{\frac{p}{2}} \Big| \hat{\boldsymbol{p}}_B\right]$$

$$= C\mathbb{E}\left[\left(\frac{p_B(s)}{m' n_0 \left(p_B(s)\lfloor \frac{n}{\lceil n_0 \hat{p}_B(s) \rceil} \rfloor \wedge 1\right)}\right)^{\frac{p}{2}} \Big| \hat{\boldsymbol{p}}_B\right]$$

$$= C\mathbb{E}\left[\left(\frac{1}{m' n_0 \left(\frac{n}{\lceil n_0 \hat{p}_B(s) \rceil}\right)} \vee \frac{p_B(s)}{m' n_0}\right)^{\frac{p}{2}} \Big| \hat{\boldsymbol{p}}_B\right]$$

$$= C\mathbb{E}\left[\left(\frac{n_0 \hat{p}_B(s) \vee 1}{m' n n_0} \vee \frac{p_B(s)}{m' n_0}\right)^{\frac{p}{2}} \Big| \hat{\boldsymbol{p}}_B\right],$$

completing the proof. $\qquad\square$

In both cases, we can take the expectation and obtain that

$$\mathbb{E}[p_B(s)^{\frac{p}{2}} \hat{p}_B(s)^{\frac{p}{2}} \|\hat{\boldsymbol{p}}_s - \boldsymbol{p}_s\|_p^p] \leq C'\mathbb{E}\left[\left(\frac{\hat{p}_B(s)}{m'n} \vee \frac{1}{m' n n_0} \vee \frac{p_B(s)}{m' n_0}\right)^{\frac{p}{2}}\right],$$

for some $C' > 0$.

Finally, take the sum over $s$ and note that $p \geq 2$, then

$$\sum_{s=1}^{t} \mathbb{E}[p_B(s)^{\frac{p}{2}} \hat{p}_B(s)^{\frac{p}{2}} \|\hat{\boldsymbol{p}}_s - \boldsymbol{p}_s\|_p^p]$$

$$\leq C' \sum_{s=1}^{t} \mathbb{E}\left[\left(\frac{\hat{p}_B(s)}{m'n} \vee \frac{1}{m' n n_0} \vee \frac{p_B(s)}{m' n_0}\right)^{\frac{p}{2}}\right]$$

$$= O\left(\left(\frac{1}{m' n_0}\right)^{\frac{p}{2}} \vee \frac{t}{(m' n n_0)^{\frac{p}{2}}}\right)$$

$$= O\left(\frac{\left(1 \vee \frac{t}{n^{\frac{p}{2}}}\right) \cdot \left(\frac{l_0}{l} \vee \frac{1}{n}\right)^{\frac{p}{2}}}{m'^{\frac{p}{2}}}\right),$$

which completes the proof.

## B.5 Proof of Proposition 2: Analysis of The Protocol for Case 2

By the case 1, the estimation error for the reduced block distribution is bounded by

$$C_3 \cdot \left[\left(\frac{t}{mnl}\right)^{\frac{p}{2}} \vee \frac{1}{(mn)^{\frac{p}{2}}}\right]$$

for some $C_3 > 0$.

By Lemma 6, the estimation error for the conditional distribution induced by the invoking of the subroutine $\text{SSRSub}(\frac{m}{2}, n, k, l, l_0, p)$ is bounded by

$$C_4 \cdot \left(\frac{\left(\frac{l_0}{l} \vee \frac{1}{n}\right)}{\frac{m}{2}}\right)^{\frac{p}{2}} = C_4 \left(\frac{2\left(\frac{l_0}{l} \vee \frac{1}{n}\right)}{m}\right)^{\frac{p}{2}} = C_4 \left[\left(\frac{2 l_0}{ml}\right)^{\frac{p}{2}} \vee \frac{2^{\frac{p}{2}}}{(mn)^{\frac{p}{2}}}\right]$$

$$\leq C_4 \left[\left(\frac{4 \log(\frac{k}{n} + 1)}{ml}\right)^{\frac{p}{2}} \vee \frac{2^{\frac{p}{2}}}{(mn)^{\frac{p}{2}}}\right],$$

for some $C_4 > 0$.

Then by Lemma 5, the total error is bounded by

$$2^{p-1}\left\{C_4\cdot\left[\left(\frac{4\log(\frac{k}{n}+1)}{ml}\right)^{\frac{p}{2}}\vee\frac{2^{\frac{p}{2}}}{(mn)^{\frac{p}{2}}}\right]+C_3\cdot\left[\left(\frac{t}{mnl}\right)^{\frac{p}{2}}\vee\frac{1}{(mn)^{\frac{p}{2}}}\right]\right\}$$

$$=O\left(\left(\frac{\log\left(\frac{k}{n}+1\right)}{ml}\right)^{\frac{p}{2}}\vee\frac{1}{(mn)^{\frac{p}{2}}}\right).$$

## B.6 Proof of Proposition 2: Analysis of The Protocol for Case 3

By the analysis in Appendix B.2.2, the estimation error for the reduced block distribution induced by the invocation of $\mathrm{SSR}(\frac{m}{2}, n, k_{a+1}, l, p)$ is bounded by

$$C_5\cdot\left[\left(\frac{\log\left(\frac{k_{a+1}}{n}+1\right)}{ml}\right)^{\frac{p}{2}}\vee\frac{1}{(mn)^{\frac{p}{2}}}\right]\leq\frac{C_5}{m^{\frac{p}{2}}},$$

for some $C_5 > 0$.

We have $k_{u+1} \geq k_{a+1} > n$ and $\frac{l_0}{l} = 1 > \frac{1}{n}$. Then by Lemma 6, the estimation error for the conditional distribution induced by the $u$-th invocation of the subroutine $\mathrm{SSRSub}(m_u, n, k_u, l, l_0, p)$ is bounded by

$$C_6\cdot\left(\frac{k_{u+1}}{m_u n}\right)^{\frac{p}{2}}\leq C_6\left(\frac{\frac{k}{2^{u(l-1)}}}{\frac{m}{2^{u+2}}n}\right)^{\frac{p}{2}}=C_6\left(\frac{2^{u+2}k}{(2^{l-1})^u mn}\right)^{\frac{p}{2}}, \tag{18}$$

for some $C_6 > 0$.

Then by Lemma 5 and $l \geq 4$, the total error is bounded by

$$2^{a(p-1)}\cdot\frac{C_5}{m^{\frac{p}{2}}}+C_6\sum_{u=1}^{a}2^{u(p-1)}\cdot\left(\frac{2^{u+2}k}{(2^{l-1})^u mn}\right)^{\frac{p}{2}}$$

$$\leq 2\left(\frac{k}{n(2^l-1)}\right)^{\frac{2(p-1)}{l}}\cdot\frac{C_5}{m^{\frac{p}{2}}}+2^{3p}C_6\left(\frac{k}{2^l mn}\right)^{\frac{p}{2}}$$

$$=O\left(\left(\frac{k}{2^l mn}\right)^{\frac{p}{2}}\right).$$

## C The Non-interactive Protocol for the TV Loss and Its Analysis

Consider the estimation problem under the TV loss, i.e. $p = 1$. In this section, we show that a uniform budget allocation plan is sufficient in this case, thanks to the error bound (12). The advantage of the uniform allocation plan is obvious, since there is no need for the decoder to send any message to the encoders. Hence a non-interactive protocol is immediate induced, only by changing (13) to

$$r(s)=\frac{1}{t} \tag{19}$$

in the successive refinement subroutine $\mathrm{SSRSub}(m', n, k, l, l_0, 1)$ in Appendix B.1.

For simplicity, we slightly abuse the notations $\mathrm{SSRSub}(m', n, k, l, l_0, 1)$ and $\mathrm{SSR}(m, n, k, l, 1)$ to still denote the resulting non-interactive protocols. The non-interactive successive refinement subroutine $\mathrm{SSRSub}(m', n, k, l, l_0, 1)$ is presented in Algorithm 2 for completeness, where differences with Algorithm 1 are underlined.

To show Theorem 2, it remains to show the error bound in the following proposition.

**Proposition 3.** *For any $p_W \in \Delta_W$, the non-interactive protocol $\mathrm{SSR}(m, n, k, l, 1)$ outputs an estimate $\hat{p}_W$ satisfying,*

**Algorithm 2** Non-Interactive Successive Refinement Subroutine $\mathrm{SSRSub}(m', n, k, l, l_0, 1)$

**Input:** Parameters $(m', n, k, l, l_0)$, an estimate $\hat{\boldsymbol{p}}_B$ of the block distribution $\boldsymbol{p}_B$ (only at the decoder side).

**Output:** An estimate $\hat{\boldsymbol{p}}_W$ of the original distribution $\boldsymbol{p}_W$.

**Allocating frames to blocks:**

1: $n_0 \leftarrow \lfloor \frac{l}{l_0} \rfloor \wedge n$.
2: Divide each $l$-bit message into $n_0$ frames of length $l_0$.
3: **for** $s = 1 : t$ **do**
4:    $r(s) \leftarrow 1/t$.
5:    $N_s \leftarrow \lfloor m' n_0 r(s) \rfloor$.
6:    Allocate $N_s$ frames to $\mathcal{W}_s$, s.t. at most $\lceil n_0 r(s) \rceil$ frames are at the same encoder side.
7: **end for**

**Proceed as that in Algorithm 1.**

1. if $k \leq n$, $m(l \wedge k) > 1000k \log m \log n$, then $\mathbb{E}[\|\hat{\boldsymbol{p}}_W - \boldsymbol{p}_W\|_p^p] = O\left(\sqrt{\frac{k^2}{mnl}} \vee \sqrt{\frac{k}{mn}}\right)$;

2. if $n < k \leq (2^l - 1) \cdot n$, $l \geq 2$ and $m(l \wedge n) > 2000n \log m \log n$, then $\mathbb{E}[\|\hat{\boldsymbol{p}}_W - \boldsymbol{p}_W\|_p^p] = O\left(\sqrt{\frac{k \log\left(\frac{k}{n} + 1\right)}{ml}} \vee \sqrt{\frac{k}{mn}}\right)$;

3. if $k > (2^l - 1) \cdot n$, $l \geq 4$ and $m(l \wedge n) > 4000n \log m \log n$, then $\mathbb{E}[\|\hat{\boldsymbol{p}}_W - \boldsymbol{p}_W\|_p^p] = O\left(\sqrt{\frac{k^2}{2^l mn}}\right)$.

## C.1    Error Analysis of the Subroutine for $p = 1$

First, the estimation error induced by the subroutine $\mathrm{SSRSub}(m', n, k, l, l_0, 1)$ is described in the following lemma.

**Lemma 8.** *We have*

$$\sum_{s=1}^{t} \mathbb{E}[p_B(s)\|\hat{\boldsymbol{p}}_s - \boldsymbol{p}_s\|_{\mathrm{TV}}] = O\left(\sqrt{\frac{t}{m'}\left(1 \vee \frac{t}{n}\right) \cdot \left(\frac{l_0}{l} \vee \frac{1}{n}\right)}\right). \tag{20}$$

*Proof.* If $m' n_0 r(s) = \frac{m' n_0}{t} \leq 4$, since $\|\hat{\boldsymbol{p}}_s - \boldsymbol{p}_s\|_{\mathrm{TV}} \leq 2$, then

$$p_B(s)\|\hat{\boldsymbol{p}}_s - \boldsymbol{p}_s\|_{\mathrm{TV}} \leq 2p_B(s) \leq 4\sqrt{\frac{p_B(s)^2 t}{m' n_0}} \leq 4\sqrt{\frac{p_B(s)^2 k}{m' n_0}}.$$

Otherwise, we have $m' n_0 r(s) = \frac{m' n_0}{t} > 4$, hence $N_s = \Theta\left(m' n_0 r(s)\right) = \Theta\left(\frac{m' n_0}{t}\right)$. Then $\tilde{W}_u^s$ for $u = 1, ..., N_s$ are i.i.d. random variables with

$$q_s \triangleq \mathbb{P}[\tilde{W}_u^s \neq \emptyset | \hat{\boldsymbol{p}}_B] = \Theta\left(p_B(s) \left\lfloor \frac{n}{\lceil n_0 r(s) \rceil} \right\rfloor \wedge 1\right) = \Theta\left(p_B(s) \left\lfloor \frac{n}{\lceil n_0/t \rceil} \right\rfloor \wedge 1\right). \tag{21}$$

Then we can establish the following lemma.

**Lemma 9.** $\mathbb{E}[p_B(s)\|\hat{\boldsymbol{p}}_s - \boldsymbol{p}_s\|_{\mathrm{TV}}] \leq C\mathbb{E}\left[\sqrt{\frac{p_B(s)k}{m'nt} \vee \frac{p_B(s)k}{m'nn_0} \vee \frac{p_B(s)^2 k}{m'n_0}}\right]$ *for some $C > 0$.*

*Proof.* By the Chernoff bound, we have

$$\mathbb{P}\left[N_s' \geq \frac{N_s q_s}{2} \Big| \hat{\boldsymbol{p}}_B\right] \leq \exp\left(-\frac{N_s q_s}{8}\right). \tag{22}$$

And conditional on the event $\{\tilde{W}_u^s \neq \emptyset\}$, the distribution of $\tilde{W}_u^s$ is $\boldsymbol{p}_s$. By the Cauchy-Schwarz inequality and $p_s(w) \in [0,1]$,

$$\mathbb{E}\left[\|\hat{\boldsymbol{p}}_s - \boldsymbol{p}_s\|_{\mathrm{TV}} \middle| N_s' \geq \frac{N_s q_s}{2}\right] \leq \sqrt{|\mathcal{W}_s| \cdot \mathbb{E}\left[\|\hat{\boldsymbol{p}}_s - \boldsymbol{p}_s\|_2^2 \middle| N_s' \geq \frac{N_s q_s}{2}\right]} = O\left(\sqrt{\frac{|\mathcal{W}_s|}{N_s q_s}}\right).$$

Since $\|\hat{\boldsymbol{p}}_s - \boldsymbol{p}_s\|^2 \leq 2$, we have

$$\mathbb{E}[\|\hat{\boldsymbol{p}}_s - \boldsymbol{p}_s\|^2 | \hat{\boldsymbol{p}}_B] \leq 2\exp\left(-\frac{N_s q_s}{8}\right) + O\left(\sqrt{\frac{|\mathcal{W}_s|}{N_s q_s}}\right) = O\left(\sqrt{\frac{|\mathcal{W}_s|}{N_s q_s}}\right).$$

Since $n_0 \leq n$, we have $\lceil \frac{n_0}{t} \rceil \leq n$ and $\frac{n}{\lceil n_0/t \rceil} \geq 1$. Hence there exists some $C > 0$, such that

$$\mathbb{E}[p_B(s)\|\hat{\boldsymbol{p}}_s - \boldsymbol{p}_s\|_{\mathrm{TV}}] \leq C\mathbb{E}\left[\sqrt{\frac{p_B(s)^2 \frac{k}{t}}{\frac{m'n_0}{t} q_s}}\right]$$

$$= C\mathbb{E}\left[\sqrt{\frac{p_B(s)^2 k}{m'n_0 \left(p_B(s)\lfloor \frac{n}{\lceil n_0/t \rceil}\rfloor \wedge 1\right)}}\right]$$

$$= C\mathbb{E}\left[\sqrt{\frac{p_B(s)k}{m'n_0 \left(\frac{n}{\lceil n_0/t \rceil}\right)} \vee \frac{p_B(s)^2 k}{m'n_0}}\right]$$

$$= C\mathbb{E}\left[\sqrt{\frac{p_B(s)k}{m'nt} \vee \frac{p_B(s)k}{m'nn_0} \vee \frac{p_B(s)^2 k}{m'n_0}}\right],$$

completing the proof. $\qquad\square$

In both cases, we can take the expectation and obtain that

$$\mathbb{E}[p_B(s)\|\hat{\boldsymbol{p}}_s - \boldsymbol{p}_s\|_{\mathrm{TV}}] \leq C'\mathbb{E}\left[\sqrt{\frac{p_B(s)k}{m'nt} \vee \frac{p_B(s)k}{m'nn_0} \vee \frac{p_B(s)^2 k}{m'n_0}}\right],$$

for some $C' > 0$.

Finally, take the sum over $s$ and use the Cauchy-Schwarz inequality, then

$$\sum_{s=1}^{t} \mathbb{E}[p_B(s)\|\hat{\boldsymbol{p}}_s - \boldsymbol{p}_s\|_{\mathrm{TV}}]$$

$$\leq C'\sum_{s=1}^{t} \mathbb{E}\left[\sqrt{\frac{p_B(s)k}{m'nt} \vee \frac{p_B(s)k}{m'nn_0} \vee \frac{p_B(s)^2 k}{m'n_0}}\right]$$

$$= O\left(\sqrt{\frac{k}{m'n_0} \vee \frac{kt}{m'nn_0}}\right)$$

$$= O\left(\sqrt{\frac{k}{m'}\left(1 \vee \frac{t}{n}\right) \cdot \left(\frac{l_0}{l} \vee \frac{1}{n}\right)}\right),$$

which completes the proof of Proposition 3. $\qquad\square$

## C.2 Error Analysis of the Non-Interactive Protocol

We complete the proof of Proposition 3 in this subsection.

### C.2.1 Error Analysis for the Base Case 1

Since the protocol for $p = 1$ is the same as that for $p = 2$, then by the Cauchy-Schwarz inequality and the analysis in Appendix A we have

$$\mathbb{E}[\|\hat{\boldsymbol{p}}_W - \boldsymbol{p}_W\|_{\mathrm{TV}}] \leq \sqrt{k\mathbb{E}[\|\hat{\boldsymbol{p}}_W - \boldsymbol{p}_W\|_2^2]} \preceq \sqrt{\frac{k^2}{mnl}} \vee \sqrt{\frac{k}{mn}}.$$

### C.2.2 Error Analysis for Case 2

By the analysis in Appendix C.2.1, the estimation error for the reduced block distribution is bounded by

$$C_3 \cdot \left( \sqrt{\frac{t^2}{mnl}} \vee \sqrt{\frac{t}{mn}} \right),$$

for some $C_3 > 0$.

By Lemma 8, the estimation error for the conditional distribution induced by the invoking of the subroutine $\mathrm{SSRSub}(\frac{m}{2}, n, k, l, l_0, 1)$ is bounded by

$$C_4\sqrt{\frac{k\left(\frac{l_0}{l} \vee \frac{1}{n}\right)}{\frac{m}{2}}} = C_4\sqrt{\frac{2k\left(\frac{l_0}{l} \vee \frac{1}{n}\right)}{m}} = C_4\left( \sqrt{\frac{2l_0 k}{ml}} \vee \sqrt{\frac{2k}{mn}} \right)$$

$$\leq C_4 \cdot \left( \sqrt{\frac{4k\log(\frac{k}{n}+1)}{ml}} \vee \sqrt{\frac{2k}{mn}} \right),$$

for some $C_4 > 0$.

Then by (12), the total error is bounded by

$$C_3 \cdot \left( \sqrt{\frac{t^2}{mnl}} \vee \sqrt{\frac{t}{mn}} \right) + C_4 \cdot \left( \sqrt{\frac{4k\log(\frac{k}{n}+1)}{ml}} \vee \sqrt{\frac{k}{mn}} \right)$$

$$= O\left( \sqrt{\frac{k\log\left(\frac{k}{n}+1\right)}{ml}} \vee \sqrt{\frac{k}{mn}} \right).$$

### C.2.3 Error Analysis for Case 3

By the analysis in Appendix C.2.2, the estimation error for the reduced block distribution induced by the invocation of $\mathrm{SSR}(\frac{m}{2}, n, k_{a+1}, l, 1)$ is bounded by

$$C_5 \cdot \left( \sqrt{\frac{k_{a+1}\log\left(\frac{k_{a+1}}{n}+1\right)}{ml}} \vee \sqrt{\frac{k_{a+1}}{mn}} \right) \leq C_5 \cdot \sqrt{\frac{k_{a+1}}{m}},$$

for some $C_5 > 0$.

We have $k_{u+1} \geq k_{a+1} > n$ and $\frac{l_0}{l} = 1 > \frac{1}{n}$. Then by Lemma 8, the estimation error for the conditional distribution induced by the $u$-th invocation of the subroutine $\mathrm{SSRSub}(m_u, n, k_u, l, l_0, 1)$ is bounded by

$$C_6 \cdot \sqrt{\frac{k_{u+1} \cdot k_u}{m_u n}} \leq C_6\sqrt{\frac{\frac{k}{2^{u(l-1)}} \cdot k}{\frac{m}{2^{u+2}} n}} = C_6\sqrt{\frac{2^{u+2}k^2}{(2^{l-1})^u mn}}, \tag{23}$$

for some $C_6 > 0$.

Then by (12) and $l \geq 4$, the total error is bounded by

$$C_5 \cdot \sqrt{\frac{k_{a+1}}{m}} + C_6 \sum_{u=1}^{a} \sqrt{\frac{2^{u+2}k^2}{(2^{l-1})^u mn}} \leq C_5 \cdot \sqrt{\frac{k}{m}} + 8C_6\sqrt{\frac{k^2}{2^l mn}} = O\left( \sqrt{\frac{k^2}{2^l mn}} \right).$$

# D  The Protocol for the Case (5b) and Its Analysis

In this section, we design a refinement protocol with sample compression that achieves the optimal rates for the case (5b), summarized in the following proposition.

**Proposition 4.** *Let $p \geq 2$, $k > n$, $ml \geq 1000n \log(mn) \log k$ and $\lceil \log k \rceil \leq l \leq n^{\frac{2}{p}}$. Then for the problem in Section 2, there exists an interactive protocol such that for any $\boldsymbol{p}_W \in \Delta_{\mathcal{W}}$, the protocol outputs an estimate $\hat{\boldsymbol{p}}_W$ satisfying $\mathbb{E}[\|\hat{\boldsymbol{p}}_W - \boldsymbol{p}_W\|_p^p] = O\left( \frac{\log^{\frac{p}{2}} k}{(ml)^{\frac{p}{2}} n^{\frac{p}{2}-1}} \right).$*

Note that the communication budget $l \geq \lceil \log k \rceil$ is sufficient to encode more than one sample. A naive idea is to let each terminal transmit their i.i.d. samples directly, so that the decoder can infer the distribution based on the samples.

To achieve higher accuracy, a subset $\mathcal{W}'$ containing $w$ with relatively larger $p_W(w)$ is identified and those $p_W(w)$ needs to be refined. A sample compression technique projects each sample to the subset $\mathcal{W}'$, which makes the encoding of the samples efficient. The protocol designed in Appendix A is then used to refine the distribution on $\mathcal{W}'$. We present the details as follows.

## D.1  The Refinement Protocol with Sample Compression

**Transmit multiple samples**  Let $n_0 = \lfloor \frac{l}{\lceil \log k \rceil} \rfloor \leq n$. Each of the first $\frac{m}{3}$ encoders divides its $l$-bit message into $n_0$ frames, and each frame has $\lceil \log k \rceil$ bits. Then encode each of its first $n_0$ samples by one of these $n_0$ frames. Send the message to the decoder.

Receiving the message, the decoder can access $M_1 \triangleq mn_0$ i.i.d. random samples $(W_l^1)_{l=1}^{M_1}$. Then for each $w \in \mathcal{W}$, let

$$\hat{\boldsymbol{p}}_W^1(w) = \frac{\sum_{l=1}^{M_1} \mathbb{1}_{W_l^1 = w}}{M_1}$$

and output the estimate $\hat{\boldsymbol{p}}_W^1$.

**Refinement with sample compression**  Based on the estimate $\hat{\boldsymbol{p}}_W^1$, the decoder computes

$$\mathcal{W}' = \left\{ w \in \mathcal{W} : \hat{p}_W^1(w) > \frac{2}{n} \right\},$$

where it is immediate that $|\mathcal{W}'| \leq n - 1$ since $\hat{\boldsymbol{p}}_W^1$ is normalized. All the remaining $\frac{2m}{3}$ encoders are informed of $\mathcal{W}'$.

Let the second $\frac{m}{3}$ encoders and the decoder repeat the protocol in the first step, so that an estimate $\hat{\boldsymbol{p}}_W^2(w)$ is obtained by the decoder.

Finally, consider the last $\frac{m}{3}$ encoders. For the $i$-th encoder among them, it computes $W_{ij}' = h(W_{ij})$ for $j = 1, ..., n$, where $(W_{ij})_{j=1}^n$ are its observed samples and

$$h(w) = \begin{cases} w, w \in \mathcal{W}', \\ \emptyset, w \notin \mathcal{W}'. \end{cases}$$

Let $W' = h(W)$ and $p_{W'}$ be its distribution of dimension no more than $n$. Then each encoder holds $n$ i.i.d. samples $(W_{ij}')_{j=1}^n$ and $W_{ij}' \sim p_{W'}$. Let these encoders and the decoder invoke the protocol $\mathrm{IR}(\frac{m}{2}, n, |\mathcal{W}'| + 1, l, p)$ defined in Appendix A (which is possible since $|\mathcal{W}'| + 1 \leq n$ and $ml \geq 1000(|\mathcal{W}'| + 1) \log(mn) \log n$). The decoder can obtain the estimate $\hat{\boldsymbol{p}}_{W'}^3$ for $\boldsymbol{p}_{W'}$.

Finally, for each $w \in \mathcal{W}$, the decoder computes

$$\hat{p}_W^3(w) = \begin{cases} \hat{p}_{W'}^3(w), & w \in \mathcal{W}', \\ \hat{p}_W^2(w), & w \notin \mathcal{W}', \end{cases}$$

and outputs the estimate $\hat{\boldsymbol{p}}_W^3$.

## D.2 Proof of Proposition 4: Error Analysis for the Protocol in Appendix D.1

It is easy to analyze the error for the rough estimate $\hat{p}_W^1$. For each $w \in \mathcal{W}$, it is folklore that for $p \geq 1$ (cf. Theorem 4 in [22] or Rosenthal's inequality [23]),

$$\mathbb{E}[|\hat{p}_W^1(w) - p_W(w)|^p] = O\left(\left(\frac{p_W(w)}{mn_0}\right)^{\frac{p}{2}} + \mathbb{1}_{p \geq 2} \cdot \frac{p_W(w)}{(mn_0)^{p-1}}\right). \tag{24}$$

*Remark* 11 (Necessity of the refinement method). For $1 \leq p \leq 2$, taking the summation and using the Hölder's Inequality imply that

$$\mathbb{E}[\|\hat{\boldsymbol{p}}_W^1 - \boldsymbol{p}_W\|_p^p] \leq O\left(\frac{k^{1-\frac{p}{2}}}{(mn_0)^{\frac{p}{2}}}\right) = O\left(\frac{k^{1-\frac{p}{2}}\log^{\frac{p}{2}} k}{(ml)^{\frac{p}{2}}}\right).$$

The bound is tight up to logarithm factors for $1 \leq p \leq 2$. However, for $p > 2$ we can only get the total error bound $O\left(\frac{\log^{\frac{p}{2}} k}{(ml)^{\frac{p}{2}}}\right)$, which is not tight. In contrast, the refined estimate $\hat{\boldsymbol{p}}_W^3$ can achieve a better upper bound and we show $\mathbb{E}[\|\hat{\boldsymbol{p}}_W^3 - \boldsymbol{p}_W\|_p^p] = O\left(\frac{\log^{\frac{p}{2}} k}{(ml)^{\frac{p}{2}} n^{\frac{p}{2}-1}}\right)$ in the following.

To complete the proof of Proposition 4, it suffices to show that $\mathbb{E}[\|\hat{\boldsymbol{p}}_W^3 - \boldsymbol{p}_W\|_p^p] = O\left(\frac{1}{(mn_0)^{\frac{p}{2}} n^{\frac{p}{2}-1}}\right)$.

We can obtain the following preliminary results, characterizing the estimation errors for the first and the second step. The proof is derived from (24) and similar to the proof of Lemma 4: for $p_W(w) > \frac{4}{n}$,

$$\mathbb{P}\left[\hat{p}_W^1(w) \leq \frac{p_W(w)}{2}\right] = O\left(\frac{1}{(mn_0 p_W(w))^{\frac{p}{2}}}\right). \tag{25}$$

By (10) in the proof of Proposition 1, we have

$$\mathbb{E}\left[|p_W(w) - \hat{p}_{W'}^3(w)|^p | w \in \mathcal{W}'\right] = O\left(\frac{1}{(mnl)^{\frac{p}{2}}} + \left(\frac{n}{mnl}\right)^{p-1} p_W(w) + \frac{p_W(w)^{\frac{p}{2}}}{(mn)^{\frac{p}{2}}}\right). \tag{26}$$

Note that

$$\mathbb{E}[\|\hat{\boldsymbol{p}}_W^3 - \boldsymbol{p}_W\|_p^p] \leq \sum_{w:p_W(w)\leq\frac{4}{n}} \mathbb{E}\left[|p_W(w) - \hat{p}_W^3(w)|^p\right] + \sum_{w:p_W(w)>\frac{4}{n}} \mathbb{E}\left[|p_W(w) - \hat{p}_W^3(w)|^p\right].$$

It suffices to bound the above two terms separately.

If $p_W(w) \leq \frac{4}{n}$, then by the error bounds (24) (applied to $\hat{\boldsymbol{p}}_W^2$) and (26), we have

$$\begin{aligned}
&\mathbb{E}\left[|p_W(w) - \hat{p}_W^3(w)|^p\right] \\
=&\mathbb{E}\left[\mathbb{1}_{w\in\mathcal{W}'}|p_W(w) - \hat{p}_{W'}^3(w)|^p\right] + \mathbb{E}\left[\mathbb{1}_{w\notin\mathcal{W}'}|p_W(w) - \hat{p}_W^2(w)|^p\right] \\
\leq&\mathbb{P}[w \in \mathcal{W}']\mathbb{E}\left[|p_W(w) - \hat{p}_{W'}^3(w)|^p | w \in \mathcal{W}'\right] + \mathbb{E}\left[|p_W(w) - \hat{p}_W^2(w)|^p\right] \\
\leq&O\left(\frac{\mathbb{P}[w \in \mathcal{W}']}{(mnl)^{\frac{p}{2}}} + \left(\frac{1}{ml}\right)^{p-1} p_W(w) + \frac{p_W(w)^{\frac{p}{2}}}{(mn)^{\frac{p}{2}}}\right) + O\left(\left(\frac{p_W(w)}{mn_0}\right)^{\frac{p}{2}} + \frac{p_W(w)}{(mn_0)^{p-1}}\right) \\
=&O\left(\frac{\mathbb{P}[w \in \mathcal{W}']}{(mnl)^{\frac{p}{2}}} + \frac{p_W(w)}{(mn_0)^{\frac{p}{2}} n^{\frac{p}{2}-1}}\right).
\end{aligned}$$

Take the summation and note that $|\mathcal{W}'| \leq n$, then

$$\begin{aligned}
\sum_{w:p_W(w)\leq\frac{4}{n}} \mathbb{E}\left[|p_W(w) - \hat{p}_W^3(w)|^p\right] \leq&O\left(\sum_{w:p_W(w)\leq\frac{4}{n}} \frac{\mathbb{P}[w \in \mathcal{W}']}{(mnl)^{\frac{p}{2}}} + \frac{p_W(w)}{(mn_0)^{\frac{p}{2}} n^{\frac{p}{2}-1}}\right) \\
\leq&O\left(\frac{\mathbb{E}[|\mathcal{W}'|]}{(mnl)^{\frac{p}{2}}} + \frac{1}{(mn_0)^{\frac{p}{2}} n^{\frac{p}{2}-1}}\right) = O\left(\frac{1}{(mn_0)^{\frac{p}{2}} n^{\frac{p}{2}-1}}\right).
\end{aligned} \tag{27}$$

If $p_W(w) > \frac{4}{n}$, then $\mathbb{P}[w \notin \mathcal{W}'] \leq \mathbb{P}\left[\hat{p}_W^1(w) \leq \frac{p_W(w)}{2}\right]$. By (24) (applied to $\hat{\boldsymbol{p}}_W^2$), (25) and (26), we have

$$
\begin{aligned}
&\mathbb{E}\left[|p_W(w) - \hat{p}_W^3(w)|^p\right] \\
=&\mathbb{E}\left[\mathbb{1}_{w \in \mathcal{W}'}|p_W(w) - \hat{p}_{W'}^3(w)|^p\right] + \mathbb{E}\left[\mathbb{1}_{w \notin \mathcal{W}'}|p_W(w) - \hat{p}_W^2(w)|^p\right] \\
\leq&\mathbb{E}\left[|p_W(w) - \hat{p}_{W'}^3(w)|^p | w \in \mathcal{W}'\right] + \mathbb{P}[w \notin \mathcal{W}'] \cdot \mathbb{E}\left[|p_W(w) - \hat{p}_W^2(w)|^p\right] \\
\leq&O\left(\frac{1}{(mnl)^{\frac{p}{2}}} + \left(\frac{1}{ml}\right)^{p-1} p_W(w) + \frac{p_W(w)^{\frac{p}{2}}}{(mn)^{\frac{p}{2}}}\right) \\
&+O\left(\frac{1}{(mn_0 p_W(w))^{\frac{p}{2}}} \cdot \left[\left(\frac{p_W(w)}{mn_0}\right)^{\frac{p}{2}} + \frac{p_W(w)}{(mn_0)^{p-1}}\right]\right) \\
=&O\left(\frac{1}{(mnn_0)^{\frac{p}{2}}} + \left(\frac{1}{ml}\right)^{p-1} p_W(w) + \frac{p_W(w)^{\frac{p}{2}}}{(mn)^{\frac{p}{2}}}\right),
\end{aligned}
$$

where the last step is since $p_W(w) > \frac{4}{n}$ and $mn_0 \geq \frac{ml}{4\log k} > 1000n$. Take the summation and note that $|\{w : p_W(w) > \frac{4}{n}\}| \leq n$, we have

$$
\begin{aligned}
&\sum_{w:p_W(w)>\frac{4}{n}} \mathbb{E}\left[|p_W(w) - \hat{p}_W^3(w)|^p\right] \\
\leq&O\left(\sum_{w:p_W(w)>\frac{4}{n}} \frac{1}{(mnn_0)^{\frac{p}{2}}} + \left(\frac{1}{ml}\right)^{p-1} p_W(w) + \frac{p_W(w)^{\frac{p}{2}}}{(mn)^{\frac{p}{2}}}\right) \quad\quad (28) \\
=&O\left(\frac{1}{(mn_0)^{\frac{p}{2}}n^{\frac{p}{2}-1}}\right),
\end{aligned}
$$

where the last step is since $n_0 = \lfloor \frac{l}{\lceil \log k \rceil} \rfloor \leq n^{\frac{2}{p}}$. Combining (27) and (28), we complete the proof of Proposition 4.

## E    The Protocol for Cases (1d) and (5d) and Its Analysis

In this section, we design a refinement protocol with thresholding that achieves the optimal rates for cases (1d) and (5d). It suffices to prove the following proposition in this section.

**Proposition 5.** *For the problem in Section 2 and each of the following cases, there exists an interactive protocol such that for any $\boldsymbol{p}_W \in \Delta_{\mathcal{W}}$, the protocol outputs an estimate $\hat{\boldsymbol{p}}_W$ satisfying*

1. *If $1 \leq p \leq 2$, $\lceil \log k \rceil \leq l \leq n$ and $ml < k$, then $\mathbb{E}[\|\hat{\boldsymbol{p}}_W - \boldsymbol{p}_W\|_p^p] = O\left(\frac{\log^{\frac{p}{2}} k}{(ml)^{p-1}}\right)$.*

2. *If $p > 2$, $\lceil \log k \rceil \leq l \leq n$ and $ml < n$, then $\mathbb{E}[\|\hat{\boldsymbol{p}}_W - \boldsymbol{p}_W\|_p^p] = O\left(\frac{\log^{p-1} k \vee \log^{2p-1}(mn) \log^{2p-1} n}{(ml)^{p-1}} \vee \frac{1}{(mn)^{\frac{p}{2}}}\right)$.*

To overcome the difficulty induced by the extremely tight total communication budget, huge "preys" and little "flies" among all $p_W(w)$ to be estimated should be classified and dealt with differently. The thresholding level is naturally $\frac{1}{ml}$, since roughly $\sim ml$ samples can be transmitted by the first step of the protocol in Appendix D.1. For those little "flies" $p_W(w) \preceq \frac{1}{ml}$, it is better to overlooking them than trying to estimate them. The remaining budgets should be used for refining huge "preys" $p_W(w) \succeq \frac{1}{ml}$ whose number $\sim ml$ is limited, by generating another independent estimate. For $p > 2$, sample compression strategies and the protocol in Appendix A are applied to refine the estimate similar to the refinement step of the protocol in Appendix D.1. With the help of thresholding, the resulting estimation protocol can catch the rough landscape of the distribution $\boldsymbol{p}_W$ and achieve the optimal error rate under the communication constraints.

We present the protocols for two cases respectively in the following subsections and detailed error analysis can be found in Appendices E.3 and E.4.

### E.1 Thresholding Methods for Case 1

**Rough estimation** Let $n_0 = \lfloor \frac{l}{\lceil \log k \rceil} \rfloor \leq n$. Let the first $\frac{m}{2}$ encoders and the decoder invoke the first step (namely the "transmit multiple sample" step) of the protocol presented in Appendix D.1, so that the decoder can obtain an estimate $\hat{\boldsymbol{p}}_W^1$.

**Thresholding technique** Based on that, the decoder computes

$$\mathcal{W}' = \left\{ w \in \mathcal{W} : \hat{p}_W^1(w) > \frac{2}{ml} \right\},$$

where it is immediate that $|\mathcal{W}'| \leq ml$ since $\hat{\boldsymbol{p}}_W^1$ is normalized.

Let the second $\frac{m}{2}$ encoders and the decoder repeat the first step of the protocol in Appendix D.1, so that an estimate $\hat{\boldsymbol{p}}_W^2(w)$ is obtained by the decoder.

Then for each $w \in \mathcal{W}$, the decoder computes

$$\hat{p}_W^3(w) = \begin{cases} \hat{p}_W^2(w), & w \in \mathcal{W}', \\ 0, & w \notin \mathcal{W}', \end{cases}$$

and outputs the estimate $\hat{\boldsymbol{p}}_W^3$.

### E.2 Combining Thresholding Methods and Refinement for Case 2

**Rough estimation** Let $k' = \frac{ml}{2000 \log(mn) \log n}$, then $k' < ml < n$ and $ml > 1000k' \log(mn) \log n$.

Let the first $\frac{m}{2}$ encoders and the decoder invoke the protocol presented in the first step of Appendix D.1. Then the decoder can obtain an estimate $\hat{\boldsymbol{p}}_W^1$.

**The mixed thresholding and refinement technique** Based on that, the decoder computes

$$\mathcal{W}' = \left\{ w \in \mathcal{W} : \hat{p}_W^1(w) > \frac{2}{k'} \right\},$$

where it is immediate that $|\mathcal{W}'| \leq k' - 1$ since $\hat{\boldsymbol{p}}_W^1$ is normalized. All the remaining $\frac{m}{2}$ encoders are informed of $\mathcal{W}'$.

Then consider the second $\frac{m}{2}$ encoders. For the $i$-th encoder among them, it computes $W_{ij}' = h(W_{ij})$ for $j = 1, ..., n$, where $(W_{ij})_{j=1}^n$ are its observed samples and

$$h(w) = \begin{cases} w, w \in \mathcal{W}', \\ \emptyset, w \notin \mathcal{W}'. \end{cases}$$

Let $W' = h(W)$ and $p_{W'}$ be its distribution of dimension no more than $n$. Then each encoder holds $n$ i.i.d. samples $(W_{ij}')_{j=1}^n$ and $W_{ij}' \sim p_{W'}$. Let these encoders and the decoder invoke the protocol $\text{IR}(\frac{m}{2}, n, |\mathcal{W}'| + 1, l, p)$ defined in Appendix A (which is possible since $|\mathcal{W}'| + 1 \leq k' < n$ and $ml \geq 1000(|\mathcal{W}'| + 1) \log(mn) \log n$). The decoder can obtain the estimate $\hat{\boldsymbol{p}}_{W'}^2$ for $\boldsymbol{p}_{W'}$. Then for each $w \in \mathcal{W}$, it computes

$$\hat{p}_W^3(w) = \begin{cases} \hat{p}_{W'}^2(w), & w \in \mathcal{W}', \\ 0, & w \notin \mathcal{W}', \end{cases}$$

and outputs the estimate $\hat{\boldsymbol{p}}_W^3$.

### E.3 Error Analysis for the Protocol in Appendix E.1

It suffices to show that $\mathbb{E}[\|\hat{\boldsymbol{p}}_W^3 - \boldsymbol{p}_W\|_p^p] = O\left( \frac{1}{(mn_0)^{\frac{p}{2}} (ml)^{\frac{p}{2}-1}} \right)$.

We first give the following preliminary results, characterizing the estimation error for the first step. The proof is derived from (24), similar to the proof of Lemma 4 but simpler.

$$\mathbb{P}\left[\hat{p}_W^1(w) \leq \frac{p_W(w)}{2}\right] \leq O\left(\frac{1}{(mn_0 p_W(w))^{\frac{p}{2}}}\right). \tag{29}$$

Note that

$$\mathbb{E}[\|\hat{\boldsymbol{p}}_W^3 - \boldsymbol{p}_W\|_p^p] \leq \sum_{w:p_W(w) \leq \frac{4}{ml}} \mathbb{E}\left[|p_W(w) - \hat{p}_W^3(w)|^p\right] + \sum_{w:p_W(w) > \frac{4}{ml}} \mathbb{E}\left[|p_W(w) - \hat{p}_W^3(w)|^p\right].$$

It suffices to bound the two terms separately. If $p_W(w) \leq \frac{4}{ml}$, then by (24) (applied to $\hat{\boldsymbol{p}}_{W'}^2$),

$$\begin{aligned}
\mathbb{E}\left[|p_W(w) - \hat{p}_W^3(w)|^p\right] &= \mathbb{E}\left[\mathbb{1}_{w \in \mathcal{W}'}|p_W(w) - \hat{p}_W^2(w)|^p\right] + \mathbb{E}\left[\mathbb{1}_{w \notin \mathcal{W}'}p_W(w)^p\right] \\
&\leq \mathbb{P}[w \in \mathcal{W}'] \cdot \mathbb{E}\left[|p_W(w) - \hat{p}_W^2(w)|^p\right] + p_W(w)^p \\
&= O\left(\mathbb{P}[w \in \mathcal{W}'] \cdot \left(\frac{p_W(w)}{mn_0}\right)^{\frac{p}{2}}\right) + p_W(w)^p \\
&= O\left(\mathbb{P}[w \in \mathcal{W}'] \cdot \left(\frac{1}{m^2 n_0 l}\right)^{\frac{p}{2}} + \frac{p_W(w)}{(ml)^{p-1}}\right).
\end{aligned}$$

Take the summation and note that $|\mathcal{W}'| \leq ml$, then

$$\begin{aligned}
\sum_{w:p_W(w) \leq \frac{4}{ml}} \mathbb{E}\left[|p_W(w) - \hat{p}_W^3(w)|^p\right] &\leq O\left(\sum_{w:p_W(w) \leq \frac{4}{ml}} \mathbb{P}[w \in \mathcal{W}'] \cdot \left(\frac{1}{m^2 n_0 l}\right)^{\frac{p}{2}} + \frac{p_W(w)}{(ml)^{p-1}}\right) \\
&\leq O\left(\frac{\mathbb{E}[|\mathcal{W}'|]}{(m^2 n_0 l)^{\frac{p}{2}}} + \frac{1}{(ml)^{p-1}}\right) = O\left(\frac{1}{(mn_0)^{\frac{p}{2}}(ml)^{\frac{p}{2}-1}}\right).
\end{aligned} \tag{30}$$

If $p_W(w) > \frac{4}{ml}$, then $\mathbb{P}[w \notin \mathcal{W}'] \leq \mathbb{P}\left[\hat{p}_W^1(w) \leq \frac{p_W(w)}{2}\right]$. By (24) (applied to $\hat{p}_W^2$) and (29), we have

$$\begin{aligned}
\mathbb{E}\left[|p_W(w) - \hat{p}_W^3(w)|^p\right] &= \mathbb{E}\left[\mathbb{1}_{w \in \mathcal{W}'}|p_W(w) - \hat{p}_W^2(w)|^p\right] + \mathbb{E}\left[\mathbb{1}_{w \notin \mathcal{W}'}p_W(w)^p\right] \\
&\leq \mathbb{E}\left[|p_W(w) - \hat{p}_W^2(w)|^p\right] + \mathbb{P}[w \notin \mathcal{W}'] \cdot p_W(w)^p \\
&\leq O\left(\left(\frac{p_W(w)}{mn_0}\right)^{\frac{p}{2}}\right) + p_W(w)^p \cdot O\left(\frac{1}{(mn_0 p_W(w))^{\frac{p}{2}}}\right) \\
&= O\left(\left(\frac{p_W(w)}{mn_0}\right)^{\frac{p}{2}}\right).
\end{aligned}$$

Taking the summation and noting that $|\{w : p_W(w) > \frac{4}{ml}\}| \leq ml$, by the Hölder's inequality we have

$$\sum_{w:p_W(w) > \frac{4}{ml}} \mathbb{E}\left[|p_W(w) - \hat{p}_W^3(w)|^p\right] \leq O\left(\sum_{w:p_W(w) > \frac{4}{ml}} \left(\frac{p_W(w)}{mn_0}\right)^{\frac{p}{2}}\right) = O\left(\frac{1}{(mn_0)^{\frac{p}{2}}(ml)^{\frac{p}{2}-1}}\right). \tag{31}$$

Combining (30) and (31), we complete the proof.

### E.4  Error Analysis for the Protocol in Appendix E.2

It remains to show that $\mathbb{E}[\|\hat{\boldsymbol{p}}_W^3 - \boldsymbol{p}_W\|_p^p] = O\left(\frac{1}{k'^{p-1}} \vee \frac{(ml)^p}{(mn_0)^{2p-1}}\right).$

We first give the following preliminary results, characterizing the estimation error for the first step. The proof is derived from (24), similar to the proof of Lemma 4 (where $p$ in Lemma 4 is replaced by $2p$). For $p_W(w) > \frac{4}{k'}$,

$$\mathbb{P}\left[\hat{p}_W^1(w) \leq \frac{p_W(w)}{2}\right] \leq O\left(\frac{(ml)^{p-1}}{(mn_0)^{2p-1}(p_W(w))^p}\right). \tag{32}$$

By (10) in the proof of Proposition 1, we have

$$\mathbb{E}\left[|p_W(w) - \hat{p}_{W'}^2(w)|^p|w \in \mathcal{W}'\right] = O\left(\frac{1}{(mnl)^{\frac{p}{2}}} + \left(\frac{k'}{mnl}\right)^{p-1} p_W(w) + \frac{p_W(w)^{\frac{p}{2}}}{(mn)^{\frac{p}{2}}}\right). \quad (33)$$

Note that

$$\mathbb{E}[\|\hat{\boldsymbol{p}}_W^3 - \boldsymbol{p}_W\|_p^p] \leq \sum_{w:p_W(w)\leq\frac{4}{k'}} \mathbb{E}\left[|p_W(w) - \hat{p}_W^3(w)|^p\right] + \sum_{w:p_W(w)>\frac{4}{k'}} \mathbb{E}\left[|p_W(w) - \hat{p}_W^3(w)|^p\right].$$

It suffices to bound the two terms separately. If $p_W(w) \leq \frac{4}{k'}$, then by (33) (applied to $\hat{\boldsymbol{p}}_W^2$), we have

$$\begin{aligned}
&\mathbb{E}\left[|p_W(w) - \hat{p}_W^3(w)|^p\right] \\
=&\mathbb{E}\left[\mathbb{1}_{w\in\mathcal{W}'}|p_W(w) - \hat{p}_{W'}^2(w)|^p\right] + \mathbb{E}\left[\mathbb{1}_{w\notin\mathcal{W}'}p_W(w)^p\right] \\
\leq&\mathbb{P}[w\in\mathcal{W}']\mathbb{E}\left[|p_W(w) - \hat{p}_{W'}^2(w)|^p|w\in\mathcal{W}'\right] + p_W(w)^p \\
\leq&O\left(\mathbb{P}[w\in\mathcal{W}']\cdot\left[\frac{1}{(mnl)^{\frac{p}{2}}} + \left(\frac{k'}{mnl}\right)^{p-1}p_W(w)\right] + \frac{p_W(w)^{\frac{p}{2}}}{(mn)^{\frac{p}{2}}}\right) + O\left(\frac{p_W(w)}{k'^{p-1}}\right) \\
=&O\left(\frac{\mathbb{P}[w\in\mathcal{W}']}{(mnl)^{\frac{p}{2}}} + \frac{p_W(w)}{k'^{p-1}}\right).
\end{aligned}$$

Take the summation and note that $|\mathcal{W}'| \leq k'$, then

$$\begin{aligned}
\sum_{w:p_W(w)\leq\frac{4}{k'}} \mathbb{E}\left[|p_W(w) - \hat{p}_W^3(w)|^p\right] \leq& O\left(\sum_{w:p_W(w)\leq\frac{4}{k'}} \frac{\mathbb{P}[w\in\mathcal{W}']}{(mnl)^{\frac{p}{2}}} + \frac{p_W(w)}{k'^{p-1}}\right) \\
&\leq O\left(\frac{\mathbb{E}[|\mathcal{W}'|]}{(mnl)^{\frac{p}{2}}} + \frac{1}{k'^{p-1}}\right) = O\left(\frac{1}{k'^{p-1}}\right).
\end{aligned} \quad (34)$$

If $p_W(w) > \frac{4}{k'}$, then $\mathbb{P}[w\notin\mathcal{W}'] \leq \mathbb{P}\left[\hat{p}_W^1(w) \leq \frac{p_W(w)}{2}\right]$. By (32) and (33), we have

$$\begin{aligned}
&\mathbb{E}\left[|p_W(w) - \hat{p}_W^3(w)|^p\right] \\
=&\mathbb{E}\left[\mathbb{1}_{w\in\mathcal{W}'}|p_W(w) - \hat{p}_{W'}^2(w)|^p\right] + \mathbb{E}\left[\mathbb{1}_{w\notin\mathcal{W}'}p_W(w)^p\right] \\
\leq&\mathbb{E}\left[|p_W(w) - \hat{p}_{W'}^2(w)|^p|w\in\mathcal{W}'\right] + \mathbb{P}[w\notin\mathcal{W}']\cdot p_W(w)^p \\
\leq&O\left(\frac{1}{(mnl)^{\frac{p}{2}}} + \left(\frac{k'}{mnl}\right)^{p-1}p_W(w) + \frac{p_W(w)^{\frac{p}{2}}}{(mn)^{\frac{p}{2}}}\right) + O\left(\frac{(ml)^{p-1}}{(mn_0)^{2p-1}(p_W(w))^p}\cdot p_W(w)^p\right) \\
=&O\left(\frac{(ml)^{p-1}}{(mn_0)^{2p-1}} + \left(\frac{k'}{mnl}\right)^{p-1}p_W(w) + \frac{p_W(w)^{\frac{p}{2}}}{(mn)^{\frac{p}{2}}}\right),
\end{aligned}$$

where the last step is since $mn_0 = m\lfloor\frac{l}{\lceil\log k\rceil}\rfloor < ml < n$. Take the summation and note that $|\{w : p_W(w) > \frac{4}{k'}\}| \leq k' < ml$, we have

$$\begin{aligned}
&\sum_{w:p_W(w)>\frac{4}{k'}} \mathbb{E}\left[|p_W(w) - \hat{p}_W^3(w)|^p\right] \\
&\leq O\left(\sum_{w:p_W(w)>\frac{4}{k'}} \frac{(ml)^{p-1}}{(mn_0)^{2p-1}} + \left(\frac{k'}{mnl}\right)^{p-1}p_W(w) + \frac{p_W(w)^{\frac{p}{2}}}{(mn)^{\frac{p}{2}}}\right) \\
&= O\left(\frac{(ml)^p}{(mn_0)^{2p-1}} \vee \frac{1}{(mn)^{\frac{p}{2}}}\right).
\end{aligned} \quad (35)$$

Combining (34) and (35), we complete the proof.

# F  The Protocol for $n = 1$, $p \geq 2$ and Its Analysis

In this section, we design a non-interactive protocol based on random hashing, which achieves the optimal rate for $n = 1$. Similar to the discussion in Remark 7, it suffices to show the following proposition for $p \geq 2$.

**Proposition 6.** *Let $p \geq 2$, $n = 1$ and $m2^l \geq k^2$. Then there exists a non-interactive protocol such that for any $\boldsymbol{p}_W \in \Delta_W$, the protocol outputs an estimate $\hat{\boldsymbol{p}}_W$ satisfying $\mathbb{E}[\|\hat{\boldsymbol{p}}_W - \boldsymbol{p}_W\|_p^p] = O\left(\frac{k}{(m2^l)^{\frac{p}{2}}} \vee \frac{1}{m^{\frac{p}{2}}}\right)$.*

## F.1  Motivation of the Protocol

The most natural idea is to first invoke the simulation protocol in [18] to output $M = O(\frac{m2^l}{k})$ samples from the distribution $\boldsymbol{p}_W$ at the decoder side; then estimate $\boldsymbol{p}_W$ using $M$ samples by a traditional central estimation method. It can achieve the optimal minimax rate $\frac{k}{m2^l}$ for $p = 2$, and hence the optimal rate $\frac{k}{(m2^l)^{\frac{p}{2}}}$ for $1 \leq p \leq 2$. However, for $p \geq 2$, using $M$ i.i.d. samples to estimate the underlying distribution under the $\ell^p$ loss can only achieve a rate of $\frac{1}{M^{\frac{p}{2}}} = (\frac{k}{m2^l})^{\frac{p}{2}}$, which leaves a gap with the lower bound $\frac{k}{(m2^l)^{\frac{p}{2}}}$ by Lemma 1. The above naive protocol is not optimal and we can show that the lower bound $\frac{k}{(m2^l)^{\frac{p}{2}}}$ is optimal.

The subtle difference is that the minimax optimal rate without the communication constraint is $\frac{1}{M^{\frac{p}{2}}}$ for $p \geq 2$ (cf. Lemma 11), in contrast with the optimal rate $\frac{k^{1-\frac{p}{2}}}{M^{\frac{p}{2}}}$ for $1 \leq p \leq 2$. The difference was ignored by the proof of upper bound in some previous work [17], hence the optimal rate claimed therein is not true. Constructing the order-optimal protocol really deserves special care, which is the main goal in the remaining part of this section.

The aforementioned difficulty in estimation under $\ell^p$ losses can be overcome, by using a random hash function to compress the sample first, and then constructing and rescaling the histogram to obtain the estimate. No simulation step as in [18] is needed. Moreover, it is worth mentioning that the resulting protocol is non-interactive. The idea is similar to the second estimation stage in [10] for estimating a sparse distribution under communication constraints. Details of the protocol are presented in Appendix F.2, and the error analysis can be found in Appendix F.3.

## F.2  The Non-interactive Protocol Based on Random Hashing for $n = 1$

Note that it suffices to design the protocol for $2^l \leq k^{\frac{2}{p}}$.

**Encoding**  Let the $i$-th encoder generate a random hash function $h_i : \mathcal{W} \to \{0, 1\}^l$, $i = 1, ..., m$ by shared randomness (i.e. $(h_i(w))_{w \in \mathcal{W}}$ are independent and $\mathbb{P}[h_i(w) = b] = 2^{-l}$ for each $w \in \mathcal{W}$ and $b \in \{0, 1\}^l$), so that the decoder can also generate $h_i$. Observing its sample $W_i$, the $i$-th encoder computes $B_i = h_i(W_i)$ and sends it to the decoder.

**Decoding**  Upon receiving $B_i$, the decoder then computes

$$\hat{p}_W(w) = \frac{2^l}{2^l - 1} \cdot \frac{\sum_{i=1}^{m} \mathbb{1}_{h_i(w) = B_i}}{m} - \frac{1}{2^l - 1} \tag{36}$$

for each $w \in \mathcal{W}$ and outputs $\hat{\boldsymbol{p}}_W$.

## F.3  Proof of Proposition 6: Error Analysis for the Protocol in Appendix F.2

We can analyze the error of the estimate $\hat{\boldsymbol{p}}_W$ as follows. Note that for each $w \in \mathcal{W}$ and $i = 1, ..., m$,

$$\mathbb{P}[h_i(w) = B_i] = p_W(w) + \frac{1}{2^l}(1 - p_W(w)).$$

It is folklore that (cf. Theorem 4 in [22] or Rosenthal's inequality [23]),

$$\mathbb{E}\left[\left|\frac{\sum_{i=1}^{m}\mathbb{1}_{h_i(w)=B_i}}{m} - \mathbb{P}[h_1(w)=B_1]\right|^p\right]$$

$$=O\left(\left(\frac{\mathbb{P}[h_1(w)=B_1]}{m}\right)^{\frac{p}{2}} + \frac{\mathbb{P}[h_1(w)=B_1]}{m^{p-1}}\right)$$

$$=O\left(\left(\frac{p_W(w)\vee\frac{1}{2^l}}{m}\right)^{\frac{p}{2}} + \frac{p_W(w)\vee\frac{1}{2^l}}{m^{p-1}}\right).$$

Then by (36), we have

$$\mathbb{E}[|\hat{p}_W(w) - p_W(w)|^p] = O\left(\left(\frac{p_W(w)\vee\frac{1}{2^l}}{m}\right)^{\frac{p}{2}} + +\frac{p_W(w)\vee\frac{1}{2^l}}{m^{p-1}}\right)$$

as well. Note that $m2^l \geq k^2$ and $2^l \leq k^{\frac{2}{p}} \leq k$ implies that $m \geq 2^l$. By taking the summation over all $w \in \mathcal{W}$, we complete the proof of Proposition 6.

# G  Proof of Lower Bounds

In order to prove Lemmas 1 and 2, we first reorganize the lower bounds into the following three lemmas.

**Lemma 10.** *For* $1 \leq p \leq 2$, *we have*

$$R(m,n,k,l,p) \succeq \begin{cases} \dfrac{k}{(mnl)^{\frac{p}{2}}}, & n \geq k\log k, m > \left(\dfrac{k}{l}\right)^2, l \leq k, \\[3mm] \dfrac{k^{1-\frac{p}{2}}}{(ml\log k)^{\frac{p}{2}}}, & n < k\log k, m > \left(\dfrac{k}{l}\right)^2, l \leq \dfrac{n}{\log k}, \\[3mm] \dfrac{k}{(mn2^l)^{\frac{p}{2}}}, & mn2^l > k^2. \end{cases}$$

*For* $p \geq 2$, *we have*

$$R(m,n,k,l,p) \succeq \begin{cases} \dfrac{k}{(mnl)^{\frac{p}{2}}}, & n \geq k\log k, m > \left(\dfrac{k}{l}\right)^2, l \leq k^{\frac{2}{p}}, \\[3mm] \dfrac{1}{(ml)^{\frac{p}{2}}n^{\frac{p}{2}-1}\log n}, & n < k\log k, m > \left(\dfrac{n/\log n}{l}\right)^2, l \leq \left(\dfrac{n}{\log n}\right)^{\frac{2}{p}}, \\[3mm] \dfrac{k}{(mn2^l)^{\frac{p}{2}}}, & mn2^l > k^2. \end{cases}$$

**Lemma 11.** *For* $1 \leq p \leq 2$, $R(m,n,k,l,p) \succeq \frac{k^{1-\frac{p}{2}}}{(mn)^{\frac{p}{2}}}$. *For* $p \geq 2$, *then* $R(m,n,k,l,p) \succeq \frac{1}{(mn)^{\frac{p}{2}}}$.

**Lemma 12.** *If* $2ml < k$, *then* $R(m,n,k,l,p) \succeq \frac{1}{(ml)^{p-1}}$.

We show Lemmas 11 and 12 in Appendices G.1 and G.2, respectively. Then Lemma 10 is proved by exploiting the results for $p = 1$ in [14], and details can be found in Appendix G.3.

## G.1  Proof of Lemma 11

The results for $1 \leq p \leq 2$ are well-known [16, 17], hence we only give the proof for $p \geq 2$. We use the information-theoretic methods.

### G.1.1 Choose a prior distribution and lower bound the minimax risk by the Bayes risk

We can assume that $\mathcal{W} = [1 : k]$ without loss of generality. Let

$$
\begin{aligned}
\boldsymbol{p}_W^1 &= \left( \frac{1+\epsilon}{2}, \frac{1-\epsilon}{2}, 0, ..., 0 \right), \\
\boldsymbol{p}_W^2 &= \left( \frac{1-\epsilon}{2}, \frac{1+\epsilon}{2}, 0, ..., 0 \right).
\end{aligned}
\tag{37}
$$

Let $Z \sim \mathrm{Bern}(\frac{1}{2})$ and define the prior distribution to be $\boldsymbol{p}_W^Z$. Let $\mathcal{P}$ be an $(m, n, l)$-protocol defined in Section 2, then we have

$$
\begin{aligned}
\sup_{\boldsymbol{p}_W \in \Delta_{\mathcal{W}}} \mathbb{E}[\|\hat{\boldsymbol{p}}_W^{\mathcal{P}} - \boldsymbol{p}_W\|_p^p] &\geq \frac{1}{2} \left( \mathbb{E}[\|\hat{\boldsymbol{p}}_W^{\mathcal{P}} - \boldsymbol{p}_W^1\|_p^p] + \mathbb{E}[\|\hat{\boldsymbol{p}}_W^{\mathcal{P}} - \boldsymbol{p}_W^2\|_p^p] \right) \\
&= \mathbb{E}[\|\hat{\boldsymbol{p}}_W^{\mathcal{P}} - \boldsymbol{p}_W^Z\|_p^p].
\end{aligned}
$$

### G.1.2 Convert the estimation problem into a testing problem

Let

$$
\hat{Z} = \arg\min_{z \in \{0,1\}} \|\boldsymbol{p}_W^z - \hat{\boldsymbol{p}}_W^{\mathcal{P}}\|_p.
$$

Then we have

$$
\begin{aligned}
\|\boldsymbol{p}_W^{\hat{Z}} - \boldsymbol{p}_W^Z\|_p &\leq \|\hat{\boldsymbol{p}}_W^{\mathcal{P}} - \boldsymbol{p}_W^{\hat{Z}}\|_p + \|\hat{\boldsymbol{p}}_W^{\mathcal{P}} - \boldsymbol{p}_W^Z\|_p \\
&\leq 2\|\hat{\boldsymbol{p}}_W^{\mathcal{P}} - \boldsymbol{p}_W^Z\|_p.
\end{aligned}
$$

Hence we have

$$
\begin{aligned}
\mathbb{E}[\|\hat{\boldsymbol{p}}_W^{\mathcal{P}} - \boldsymbol{p}_W^Z\|_p^p] &\geq \frac{1}{2^p} \mathbb{E}[\|\boldsymbol{p}_W^{\hat{Z}} - \boldsymbol{p}_W^Z\|_p^p] \\
&= \frac{1}{2^{p-1}} \epsilon^p \mathbb{P}[\hat{Z} \neq Z].
\end{aligned}
\tag{38}
$$

Since $Z - W^{mn} - B^m - \hat{Z}$ is a Markov chain, then by the Fano's inequality, we have

$$
I(Z; B^m) \geq 1 - h\left( \mathbb{P}[\hat{Z} \neq Z] \right),
\tag{39}
$$

where $h(p) = -p \log_2 p - (1-p) \log_2(1-p)$ is the binary entropy function. If we can show that for a suitably chosen $\epsilon$,

$$
I(Z; B^m) \leq \frac{1}{2},
\tag{40}
$$

then by (38) and (39) we have

$$
\mathbb{P}[\hat{Z} \neq Z] \geq \frac{1}{10},
$$

thus

$$
\mathbb{E}[\|\hat{\boldsymbol{p}}_W^{\mathcal{P}} - \boldsymbol{p}_W^Z\|_2^2] \succeq \epsilon^p.
$$

Then we have $R(m, n, l, r) \succeq \epsilon^p$.

### G.1.3 Choose a suitable parameter

By the Markov chain $Z_s - W^{mn} - B^m$ and the data processing inequality, we have

$$
\begin{aligned}
I(Z; B^m) &\leq I(Z; W^{mn}) \\
&= \frac{1}{2} D_{\mathcal{W}^{mn}} \left( p_W^1(w^{mn}) \Big|\Big| \frac{1}{2} \left( p_W^1(w^{mn}) + p_W^2(w^{mn}) \right) \right) \\
&\quad + \frac{1}{2} D_{\mathcal{W}^{mn}} \left( p_W^2(w^{mn}) \Big|\Big| \frac{1}{2} \left( p_W^1(w^{mn}) + p_W^2(w^{mn}) \right) \right) \\
&\leq \frac{1}{4} \left( D_{\mathcal{W}^{mn}} \left( p_W^1(w^{mn}) || p_W^2(w^{mn}) \right) + D_{\mathcal{W}^{mn}} \left( p_W^2(w^{mn}) || p_W^1(w^{mn}) \right) \right) \\
&= \frac{mn}{2} D_{\mathcal{W}} \left( p_W^2(w) || p_W^1(w) \right) \\
&= \frac{mn\epsilon}{2} \log \left( 1 + \frac{2\epsilon}{1 - \epsilon} \right) \\
&\leq \frac{mn\epsilon^2}{1 - \epsilon},
\end{aligned}
$$

where the first inequality is due to the convexity of KL divergence and the second is by the fact that $\log(1 + x) \leq x$ for $x > 0$. By letting $\epsilon = (100mn)^{-\frac{1}{2}}$ we obtain that $R(m, n, l, r) \succeq (mn)^{-\frac{p}{2}}$.

### G.2 Proof of Lemma 12

The case for $ml < k$ is not hard, but it has not been fully explored in previous literature. First note that by the Hölder's inequality, we have

$$
\| \hat{p}_W - p_W \|_{\mathrm{TV}} \leq k^{1 - \frac{1}{p}} \| \hat{p}_W - p_W \|_p.
$$

Hence we have

$$
R(m, n, k, l, p) \geq k^{1-p} R(m, n, k, l, 1)^p, \tag{41}
$$

and the minimax lower bound for $p \geq 1$ is easily implied by that for $p = 1$.

We have the following folklore lemma for $p = 1$, which can be proved by the Fano's method and the data processing inequality.

**Lemma 13.** *If $2ml \leq k$, then we have $R(m, n, k, l, 1) \succeq 1$.*

Combining Lemma 13 and (41), for any $k \geq 2ml$ we have

$$
R(m, n, k, l, p) \succeq \frac{1}{k^{p-1}}.
$$

Hence we further have

$$
R(m, n, k, l, p) \geq R(m, n, 2ml, l, p) \succeq \frac{1}{(ml)^{p-1}}.
$$

### G.3 Proof of Lemma 10

For $p = 1$, we have the following lemma in [14].

**Lemma 14** ([14], Theorem 1.1 & 1.3). *1) For $n \geq k \log k$ and $m > \left( \frac{k}{l} \right)^2$, $R(m, n, k, l, 1) \succeq \sqrt{\frac{k^2}{mnl}} \wedge 1$.*

*2) For $n \leq k \log k$ and $m > \left( \frac{k}{l} \right)^2$, $R(m, n, k, l, 1) \succeq \sqrt{\frac{k}{ml \log k}} \wedge 1$.*

*3) We always have $R(m, n, k, l, 1) \succeq \sqrt{\frac{k^2}{mn2^l}} \wedge 1$.*

With the help of (41), the following three bounds is derived from three cases in Lemma 14 respectively.

**Proof of the first bound**  For $n \geq k \log k$ and $m > (\frac{k}{l})^2$ and $l \leq k$, we can obtain that $m > \frac{k}{l}$ and $mnl \geq k^2$. Then by 1) in Lemma 14 and (41),

$$R(m,n,k,l,p) \succeq \frac{k}{(mnl)^{\frac{p}{2}}}.$$

**Proof of the second bound**  If $m > (\frac{k}{l})^2$ and $l \leq k$, then $ml \log k \geq k$. Then by 2) in Lemma 14 and (41) we have

$$R(m,n,k,l,p) \succeq \frac{k^{1-\frac{p}{2}}}{(ml \log k)^{\frac{p}{2}}}.$$

Now let $p \geq 2$. Since $n \leq k \log k$ we have $k \geq \frac{n}{\log n}$. We further have

$$R(m,n,k,l,p) \geq R(m,n,\lceil n/\log n\rceil,l,p) \succeq \frac{1}{(ml)^{\frac{p}{2}} n^{\frac{p}{2}-1} \log n}.$$

as long as $m > (\frac{\lceil n/\log n\rceil}{l})^2$ and $l \leq \lceil n/\log n\rceil$.

**Proof of the third bound**  If $mn2^l \geq k^2$, then by 1) in Lemma 14 and (41) we have

$$R(m,n,k,l,p) \succeq \frac{k}{(mn2^l)^{\frac{p}{2}}}.$$

