# OpenReview forum: "Refinement Methods for Distributed Distribution Estimation under $\ell^p$-Losses"
_NeurIPS.cc/2025/Conference — NeurIPS 2025 spotlight_

### Official Review · Reviewer_sk54 · 2025-06-24

**Clarity:** 3
**Significance:** 3
**Originality:** 3
**Rating:** 4
**Confidence:** 3

**Summary:**

The paper derives minimax optimal rates for communication-limited \ell_p estimation of discrete distributions, identifying an elbow effect at p=2 and developing adaptive protocols combining refinement methods with specialized techniques to achieve these rates.

**Questions:**

Expand the motivation and implications in the introduction, add more intuitive explanations of key concepts, and include illustrative examples where possible. This will help researchers outside this specialized area better appreciate the significance of the contributions.

Demonstrate the practical applicability of the theoretical findings through numerical experiments or case studies.

**Ethical Concerns:**

["NO or VERY MINOR ethics concerns only"]

**Final Justification:**

Based on the original manuscript and the authors' responses, I have no major concerns regarding this paper.

**Limitations:**

yes

**Paper Formatting Concerns:**

No major formatting issues are detected.

**Quality:**

3

**Strengths And Weaknesses:**

The paper is technically solid. Conclusive results are obtained for most scenarios, greatly extending and strengthening the relevant results in the literature.

The current presentation is quite terse, which may limit accessibility for readers outside this specialized field. Enhancing the exposition with additional explanatory passages would improve the paper's broader appeal.

While the theoretical contributions are substantial, incorporating empirical validation would further strengthen the work by demonstrating practical applications of the theoretical findings.

---

> ### Author Rebuttal · Authors · 2025-07-31
>
> Thanks for your careful reading and constructive suggestions. We are going to revise our paper according to your suggestions in the following aspects.
>
> -   **... Enhancing the exposition with additional explanatory passages would improve the paper's broader appeal. Expand the motivation and implications in the introduction, add more intuitive explanations of key concepts, and include illustrative examples where possible. This will help researchers outside this specialized area better appreciate the significance of the contributions.**
>
> Thanks for your suggestion. In the revised version, we will add more explanatory passages throughout the whole paper. Especially, we will expand the motivation and implications in Section 1, add more explanations about the key concepts in Sections 1 and 2, and validate our general results in several special cases to provide some intuition in Section 3 and 4.
>
> -   **While the theoretical contributions are substantial, incorporating empirical validation would further strengthen the work by demonstrating practical applications of the theoretical findings.**
>
> We will try to add some empirical validation in the revised paper and later works.

---

> > ### Comment · Reviewer_sk54 · 2025-08-05
> >
> > The authors' responses are appreciated. I have decided maintain my original score.

---

### Official Review · Reviewer_qrGV · 2025-06-30

**Clarity:** 3
**Significance:** 2
**Originality:** 2
**Rating:** 4
**Confidence:** 3

**Summary:**

The authors consider the problem of interactive communication constrained discrete distribution estimation under \ell^p loss, and they obtain minimax rates for most parameter regimes (some of these regimes were not considered in previous works). In this setup there are m  distributed encoders, each observing n i.i.d. samples from the distribution and the message generated by the previous encoder, and each transmits an L-bit message to the central decoder (observed by all subsequent encoders as well). The authors upper bound approach is to start with a rough estimate which is then being successively refined and compressed, where the refinement method depends on the specific parameter regime. The lower bounds uses a standard reduction argument to a testing problem.

**Questions:**

The lower bound uses a reduction to a binary hypothesis testing, which is a very crude reduction when dealing with distribution estimation. Yet, surprisingly, this allows you to attain a tight lower bound, up to logarithmic factors. Can you elaborate on why this is happening?

**Ethical Concerns:**

["NO or VERY MINOR ethics concerns only"]

**Final Justification:**

The authors have agreed to address my comments in the revision, and as I don't currently find any problem with their main results, I am increasing my final score.

**Limitations:**

The authors adequately addressed the limitations and potential negative societal impact of their work

**Paper Formatting Concerns:**

There are no major formatting issues in this paper.

**Quality:**

3

**Strengths And Weaknesses:**

Strengths: The paper addresses an intersting and timely distributed interactive distribution estimation, and it contributes to the literature by formulating minimax rates for a large regimes of parameters and identifying interesting trade-offs and elbow effects.
Weaknesses:
Personally, I find the paper difficult to evaluate due to some of the condensed explanations compounded with some typos, which are mainly prevalent in the appendix section. For example, in A.1, it is not stated how the refined estimate \hat{p}_W^2 is generated. In the subsequent algorithm of B.1, it is not stated how r(s)=\hat{p}_B(s) is estimated. There are some typos, such as the upper bound of remark 5 should be \ell_2 norm instead of \ell_p, and the bottom of p.37 should be P(\hat{Z}\neq Z)\geq 1/10. Also, in eq. (14) the indicator function should be equal to w rather that t.
Another concern I have is the significance of some of the results. While the trade-offs presented in the paper are novel, the proof techniques are relatively straightforward. The upper bounds are based on refining a crude estimate, where the refinement is tailored to the particular parameter range considered, and the lower bound is achieved via reduction to binary testing. I would like to see the authors elaborate on their lower bounds as it is surprising these crude reductions are tight.

---

> ### Author Rebuttal · Authors · 2025-07-31
>
> Thanks for your careful reading and detailed comments. Here we give the point-to-point responses, in order to address your concerns.
>
> ## Responses to the weaknesses and Questions:
>
> -   **Another concern I have is the significance of some of the results. While the trade-offs presented in the paper are novel, the proof techniques are relatively straightforward. The upper bounds are based on refining a crude estimate, where the refinement is tailored to the particular parameter range considered, and the lower bound is achieved via reduction to binary testing.**
>
> We want to emphasize more about the technical contributions of our work here. Most of the technical difficulties lie in the proof of the upper bound, which is summarized here. We also derive compatible lower bounds for most parameter regimes. Technical contributions in the proof of lower bounds are given in the response to the next comment.
>
> First, introduction of refinement methods to the problem here constitutes a substantial improvement. The previous protocol for the $\ell^1$ loss does not apply to $\ell^p$ losses, since its optimality depends heavily on several special properties of the $\ell^1$ loss. Even for the $\ell^2$ loss, how to design an optimal protocol remains unclear. We find that the difficulty lies in the communication budget allocation strategy, where communication budgets should be invested based on the distribution to be estimated. In order to overcome the difficulty, we introduce refinement method in designing the protocols to achieve the optimal rates. Understanding the methods is relatively easy, but coming up with it is not trivial.
>
> Second, how to make this method work for different parameter regimes requires additional ideas. Many other techniques such as successive refinement, sample compression, thresholding and random hashing are also exploited to solve the problem here, neither of which are straightforward. Showing the optimality of the induced protocol also requires solid error analysis.
>
> Finally, the estimation techniques proposed in this paper have a wide potential use in many other problems, such as the problem with privacy constraints, the pointwise estimation problem and other estimation setting with similar structural assumptions on the objects to be estimated.
>
> -   **I would like to see the authors elaborate on their lower bounds as it is surprising these crude reductions are tight. The lower bound uses a reduction to a binary hypothesis testing, which is a very crude reduction when dealing with distribution estimation. Yet, surprisingly, this allows you to attain a tight lower bound, up to logarithmic factors. Can you elaborate on why this is happening?**
>
> Thanks for your comment. We want to clarify here that the crude reduction to a binary hypothesis testing is one of the key ideas, but it does not yield the optimal rates on its own. In Section G, the overall proof of lower bounds is decomposed into Lemmas 10-12, among which two major techniques are exploited.
>
> First, in the proof of Lemmas 10 and 12, lower bounds under the $\ell^p$ loss are derived from that under the $\ell^1$ loss. Most of lower bounds under the $\ell^1$ loss are the contributions of the previous work [13]. Typically, the proof in [13] uses a reduction to a hypothesis testing problem of roughly $2^{\frac{k}{2}}$ hypotheses. A large portion of difficulty in the lower bound proof has been overcome by [13]. Based on these bounds under the $\ell^1$ loss, we derive most of the bounds under the $\ell^p$ loss using Holder Inequality. Moreover, for some regimes such as that in Lemma 12, additional algebraic tricks are exploited to strengthen the bounds, so that tight bounds are obtained.
>
> However, the derived bounds are still not tight for the case $p>2$, where the second technique is introduced to resolve the difficulty. In this work, one of the major finding is that the optimal bounds are different for $p \leq 2$ and $p>2$. For $p>2$, the centralized bound without the communication constraints in Lemma 11 is little-known but easy to show. It can be achieved by the crude reduction to a binary hypothesis testing. From a high-level perspective, this reduction to a binary hypothesis testing, combined with the reduction to multiple hypotheses testing in the proof in [13], completes the proof of lower bounds. This is why we present the proof of Lemma 11 separately and emphasize its main technique. These efforts improve the bounds derived by the first technique, so that the overall lower bound is tight.
>
> To further address your concern, we will expand the proof sketch of Lemma 1 and 2 by adding more detailed explanations.
>
> -   **Personally, I find the paper difficult to evaluate due to some of the condensed explanations compounded with some typos, which are mainly prevalent in the appendix section.**
>
> Thanks for your careful reading. Since our paper tackles the problem mathematically, we have to present these technically difficult proofs in the appendix. We will revise our paper in light of your suggestions to improve readability of the appendix. Specifically, we appreciate you indicating the typos to us and will correct them in the revised paper. However, for the first two points, besides adding more details in our paper, here we want to clarify that they do not affect the correctness of the main proofs. Our detailed responses to these points and typos are as follows.
>
> -   **In A.1, it is not stated how the refined estimate** $\hat{p}_W^2$ **is generated.**
>
> As stated in the second paragraph of Section A.1, we let $m(w)$ encoders invoke the one-bit protocol in Lemma 3 and each of them sends one bit for estimating $p_W(w)$. Upon receiving these bits, the decoder can output an estimate $\hat{p}_W^2(w)$ of $p_W(w)$ following the protocol. In Lemma 3, we introduced the one-bit protocol in [13] for the distributed estimation of the mean of a binary distribution. Following the protocol, each encoder sends one bit to the decoder, and then the decoder can output an estimate of the mean of the binary distribution. Here we apply the protocol to the estimation of each $p_W(w)$. The detailed procedure for generating $\hat{p}_W^2$ has been defined in [13]. In this work, we do not dive into details of the one-bit protocol in [13], but only exploit the error bound.
>
> We will revise the paper to emphasize that the decoder generates $\hat{p}_W^2$ following the one-bit protocol.
>
> -   **In the subsequent algorithm of B.1, it is not stated how** $r(s)=\hat{p}_B(s)$ **is estimated.**
>
> Let us clarify the basic assumption of the successive refinement subroutine in Section B.1 here. We just assume that $\hat{p}_B$ has been known and is an input of the subroutine. We explained in the last sentence of the first paragraph in Section B.1: "It receives an estimate $\hat{p}_B$ of the block distribution $p_B$ of dimension $t$, and outputs an estimate $\hat{p}_W$ of the original distribution $p_W$." In other words, the subroutine is one step in the inductive procedure. $\hat{p}_B(s)$ has been generated by the previous step or the base step, and the goal of the subroutine is to generate $\hat{p}_W$ based on $\hat{p}_B$ at the decoder side through communicating with encoders. How to construct the complete protocol by inductively using the subroutine is presented in Section B.2.
>
> We are going to add a pseudo-code description of the subroutine in Section B.1 to emphasize the input of the subroutine.
>
> -   **There are some typos, such as ...**
>
> Thanks again for your careful reading. We will correct these typos in the revised paper. Moreover, we are going to do a thorough proofreading to ensure that other typos are corrected and the presentation is improved.

---

> > ### Comment · Reviewer_qrGV · 2025-08-06
> >
> > The authors detailed rebuttal have convinced me that the phrasing, typos and lack of elaboration in some of the proofs - which the authors agreed to fix in the revision - are not indicative of a problem in their main results, and should not present an obstacle to accepting the paper. I am therefore increasing my final score.

---

### Official Review · Reviewer_ec8n · 2025-07-02

**Clarity:** 3
**Significance:** 3
**Originality:** 3
**Rating:** 5
**Confidence:** 4

**Summary:**

The paper considers discrete distribution estimation under l^p losses where each user holds multiple independent samples and provides order-wise characterization of the minimax convergence rate. Previous results have been limited to p=1,2 and often single sample per user. The paper characterizes the order-optimal convergence rate for a wide range of parameters that has remained open in the literature.

**Questions:**

Please see weaknesses for suggestions on how to improve the paper.

Question: Since the scheme for p>2 builds on the pointwise estimation scheme in [10], I wonder if the results in this paper can be further developed to achieve point-wise optimal estimation at least for p>2 but ideally in all regimes.

**Ethical Concerns:**

["NO or VERY MINOR ethics concerns only"]

**Final Justification:**

I am satisfied with authors' response. I think the contributions of the paper and the generality of the results warrant acceptance. I am updating my score accordingly.

**Limitations:**

yes

**Quality:**

3

**Strengths And Weaknesses:**

Strengths:
The paper addresses the distributed discrete distribution estimation problem across a wide range of parameter settings by developing and refining achievability schemes tailored to each subregime. The problem is practically motivated, and the schemes presented are non-trivial and sufficiently novel, despite building on prior work—for instance, [13] in the regime 1<p<2 and [10] for p>2. The paper aims to handle the general case where the rate depends on five parameters, which introduces significant complexity. I appreciated the attempts to distill high-level insights in the introduction and Section 6. For example, the intuition that block-based refinement schemes are needed when 1<p<2, while individual-entry refinements are needed for  p>2 is compelling. This aligns with the observation that estimation errors for larger probabilities become more significant as p increases. Further efforts to distill high-level insights would strengthen the paper.

Weaknesses:

Since the paper tackles the problem in full generality, the communication rate ends up depending on five parameters. As a result, it’s easy for the reader to get lost in the multitude of scaling laws across various parameter regimes. This makes the paper feel somewhat dry at times, especially when the presentation leans heavily on technical detail without sufficient guiding intuition or narrative. It would be helpful to provide more intuition about why the five regimes arise in the theorems, what each one represents, how the optimal strategies differ across them, and whether these distinctions are fundamental or largely technical. In particular, some high-level takeaways on how optimal schemes vary with respect to parameters such as p<2 vs. p>2, small n vs. large  n, or tight vs. loose communication constraints would add clarity and insight.

Additionally, the writing could benefit from further revision. Some sentences are awkwardly phrased, for instance: "The same as Theorem 2, the non-iteractive protocol in [13] is constructed for the estimation problem under the TV loss." A thorough proofreading is needed to enhance the clarity  of the presentation.

---

> ### Author Rebuttal · Authors · 2025-07-31
>
> Thanks for your careful reading and detailed suggestions. Here are our point-to-point responses to your concerns.
>
> ## Responses to the weaknesses:
>
> -   **It would be helpful to provide more intuition about why the five regimes arise in the theorems, what each one represents, how the optimal strategies differ across them, and whether these distinctions are fundamental or largely technical. ...**
>
> Thanks for your suggestion. As you say, since our paper tackles the problem in full generality, it is difficult to explain distinctions of all the regimes and their corresponding optimal protocols. We believe all the regimes discussed in the theorems are fundamental, as shown by Equations (2), (6) and Table 1. We gave some explanations in the proof sketch of Theorems 1 and 3. In light of your suggestions, we will further clarify the intuition of different protocols by revising our paper in the following.
>
> 1.  Adding a few remarks immediately after the proof sketch of Theorems 1 and 3, in order to give more high-level takeaways on how and why optimal schemes vary with respect to parameters.
>
> 2.  Adding and expanding detailed comparisons between protocols in different parameter regimes, especially in the corresponding sections in the technical appendix.
>
> -   **Additionally, the writing could benefit from further revision.**
>
> We are thankful for your advice and will thoroughly revise our paper.
>
> ## Responses to the question:
>
> -   **Since the scheme for** $p>2$ **builds on the pointwise estimation scheme in [10], I wonder if the results in this paper can be further developed to achieve point-wise optimal estimation at least for p\>2 but ideally in all regimes.**
>
> Thanks for your question, which indicates a good direction for further works. A brief discussion of the pointwise estimation problem (as well as the related work [10]) will be added in Section 7 as a potential new direction.
>
> We personally believe it is very likely to further design the point-wise optimal (up to logarithmic factors) estimation protocol for $n>1$ (at least for some parameter regimes), similar to that discussed in [10] for $n = 1$ and $1 \leq p \leq 2$. Techniques in [10] are useful for further extending the estimation methods in this work to achieve the point-wise optimality. Specifically, the protocol in Section 4 for $p>2$ is relatively easier to extend, while that in Section 3 for $1 \leq p \leq 2$ is more difficult. The idiom says, "the devil is in the details". We think that at least very careful theoretical analysis, and possibly some new techniques are still needed to fully answer this question.

---

> > ### Comment · Reviewer_ec8n · 2025-08-08
> >
> > I am satisfied with the author's response. I will update my score accordingly.

---

### Official Review · Reviewer_XUHt · 2025-07-03

**Clarity:** 2
**Significance:** 2
**Originality:** 2
**Rating:** 5
**Confidence:** 4

**Summary:**

The paper studies minimax rates for the problem of (discrete) distribution learning in the distributed setting where:
- there are $m$ terminals;
- each terminal has $n$ i.i.d. samples and uses at most $l$ bits to describe its samples;
- the error in estimation is quantified using the $\ell^p$ loss for $p \geq 1$.

A previous paper [13] studied a similar problem which was limited to the case $p = 1$ (i.e. total variation loss). As there, the schema of the protocols here is to get a rough estimate, followed by a refinement procedure. To make this schema work for general $p$ requires additional ideas, which is the main contribution of this paper.

The upper bounds on the minimax rates are given by Theorem 1 (for $1 \leq p \leq 2$) and Theorem 3 (for $p > 2$); the corresponding lower bounds are given by Lemma 1 and Lemma 2, respectively.

Theorem 2 and Theorem 4 are devoted to special cases where the estimation protocol can be made non-interactive:
- Theorem 2 gives a non-interactive protocol for the special case when $p = 1$. This is exactly the case considered in the previous work [13]. Here, the authors show that their protocol extends the range of parameters to which the guarantees apply.
- Theorem 4 gives a non-interactive protocol for the case where $n = 1$ (i.e. each terminal has $1$ sample) and $p \geq 2$. This protocol uses random hashing (and thus requires shared randomness). Further, the authors point out that the lower bounds for $p > 2$ from [16] that were claimed to be optimal are not so since the random hashing protocol achieves a performance that is strictly better.

**Questions:**

In Section 5, where a random hashing based non-interactive protocol is proposed in the case where every terminal has $1$ sample, the authors point out that the simulate and infer protocol (which requires no shared randomness) results in a suboptimal rate.
- Could the authors give a intuitive explanation of why random hashing leads to optimal rates?
- Do the authors believe that shared randomness is necessary to obtain this rate? In other words, do they think it is possible to get a lower bound for private-randomness protocols that shows a strictly worse rate than the one obtainable by shared randomness?

**Ethical Concerns:**

["NO or VERY MINOR ethics concerns only"]

**Final Justification:**

I missed the final justification box, so writing it now. I am happy with author's responses to me and the other reviewers. The main concern raised by most was about the explanations and the presentation which the authors have promised to rectify. I stand by my  original score of 5.

**Limitations:**

Yes

**Quality:**

3

**Strengths And Weaknesses:**

The overall story and the flow of the paper is clear.

As already mentioned in the summary, the main contribution of the paper is to show how to make the general idea of estimate-then-refine work for distributed distribution learning with general $\ell^p$ loss. This comes at the cost of introducing interaction among the terminals: e.g., for adaptive budget allocation; for deciding the projection set for sample compression. Even so, I believe the results in the paper demonstrate that the gap in the literature of distributed estimation with $\ell^p$ losses requires something more than just generalising the analysis of existing protocols.

Presentation wise, I would have liked a more pseudo-code style description of the protocol in terms of basic primitives. I don't think it is possible to do that in the main body of the paper because of the page limit. But perhaps this can be done in the appendix?

---

> ### Author Rebuttal · Authors · 2025-07-31
>
> Thanks for your comprehensive review and positive assessment. We are going to improve our paper according to your suggestions in the following aspects.
>
> -   **I would have liked a more pseudo-code style description of the protocol in terms of basic primitives.**
>
> Thanks for your suggestion. We are going to add more pseudo-code description in the technical appendix, in order to make the presentation of the estimation protocol more clear.
>
> -   **Could the authors give an intuitive explanation of why random hashing leads to optimal rates?**
>
> We will add a remark to give the explanation. In the following, we try to present a rough answer to this question.
>
> In our understanding, it is because random hashing implicitly adopts an optimal communication budget allocation strategy. Hence it can extract information that is most useful for estimating the underlying distribution, and ensure that the estimation errors for different entries are balanced to obtain the optimality under the $\ell^p$ loss, either for $p \leq 2$ or for $p>2$.
>
> As we have discussed in Section 5 of our paper, the estimation error for each $p_W(w)$ is $(\frac{p_W(w)}{M})^{\frac{p}{2}}$ if $M$ raw samples are available at the decoder side. That is, relatively larger entries $p_W(w)$ are typically more difficult to estimate. Hence communication budgets should be invested more into estimating these entries, especially under $\ell^p$ losses with $p>2$. In this case, the simulate-and-infer protocol uses too much communication budget to estimate the smaller entries, while fails to simulate enough samples for estimating the larger entries. Compared with the simulate-and-infer protocol, random hashing reduces estimation errors for the larger entries, despite increasing the error for the smaller entries. It turns out that random hashing obtains the optimal budget allocation, and hence achieves the optimal rate.
>
> To make it more clear, for example let $p>2$, $p_W(w_1), p_W(w_2) \asymp \frac{1}{2}$ and other $p_W(w) = 0$. To have a basic understanding, we focus on the estimation error of $p_W(w_1)$ and $p_W(w_2)$. Note that $w_1, w_2$ are unknown. The simulate-and-infer protocol simulates only $\frac{m2^l}{k}$ samples, hence the estimation error is $\sim (\frac{k}{m2^l})^{\frac{p}{2}}$. In contrast, random hashing can generate $m$ compressed samples, and reduces the estimation error for $p_W(w_1), p_W(w_2)$ to $\frac{1}{m^{\frac{p}{2}}}$. Random hashing implicitly uses more communication budgets in estimating $p_W(w_1), p_W(w_2)$, achieving an equivalent sample size of $m$ that is larger than $\frac{m2^l}{k}$. This roughly explains why random hashing obtains the optimal rates, while simulate-and-infer protocol fails to do so.
>
> -   **Do the authors believe that shared randomness is necessary to obtain this rate?**
>
> It is a very interesting question that merits further investigation. We will try to answer it in the revised paper, and later works.

---

> > ### Comment · Reviewer_XUHt · 2025-08-06
> >
> > Thanks to the authors for addressing my comments and questions. I see how random hashing might help for higher p (as opposed to, say, simulate and infer) since the problem more and more resembles sparse estimation as p grows. Proving necessity of shared randomness would be great but I don't expect the authors to do that in this paper. I have no further questions.

---

### Decision · Program_Chairs · 2025-09-17

**Decision:**

Accept (spotlight)

**Comment:**

This paper studies the problem of distributed distribution estimation under communication constraints. While this is a classical problem which has been studied extensively in recent years, this paper provides a complete picture to the general setting of

1. each node holds multiple observations;
2. a strong interactive communication protocol; and
3. a general \ell^p loss with p\ge 1.

This paper gives a complete characterization of different regimes, with matching upper and lower bounds in each regime. In particular, several different algorithmic ideas are applied in different regimes. The reviewers unanimously appreciate the merits of this paper, and I'm happily recommending acceptance. Meanwhile, the presentation of this paper can be greatly improved if some pictures of regime classifications can be drawn, and intuitions can be provided for the results in each regime.